# Human subcortical pathways automatically detect collision trajectory without attention and awareness

Fanhua Guo[1,2]ↄ, Jinyou Zou[1,2,3]ↄ, Ye Wang[1,2]ↄ, Boyan Fang[4], Huanfen Zhou[5],
Dajiang Wang[5]*, Sheng He[1,2,6]*, Peng Zhang[1,2,6]*

1 State Key Laboratory of Brain and Cognitive Science, Institute of Biophysics, Chinese Academy of Sciences, Beijing, China, 2 University of Chinese Academy of Sciences, Beijing, China, 3 Aier Institute of Optometry and Vision Science, Aier Eye Hospital Group, Changsha, China, 4 Neurological Rehabilitation Center, Beijing Rehabilitation Hospital, Capital Medical University, Beijing, China, 5 Division of Ophthalmology, The Third Medical Center of PLA General Hospital, Beijing, China, 6 Institute of Artificial Intelligence, Hefei Comprehensive National Science Center, Hefei, China

ↄ These authors contributed equally to this work.
* wangdajiang301@163.com (DW); hes@ibp.ac.cn (SH); zhangpeng@ibp.ac.cn (PZ)

**Data Availability Statement:** Data and code to reproduce the main findings of this study can be downloaded from Open Science Framework (OSF, https://doi.org/10.17605/OSF.IO/GDJWH).

## Abstract

Detecting imminent collisions is essential for survival. Here, we used high-resolution fMRI at 7 Tesla to investigate the role of attention and consciousness for detecting collision trajectory in human subcortical pathways. Healthy participants can precisely discriminate collision from near-miss trajectory of an approaching object, with pupil size change reflecting collision sensitivity. Subcortical pathways from the superior colliculus (SC) to the ventromedial pulvinar (vmPul) and ventral tegmental area (VTA) exhibited collision-sensitive responses even when participants were not paying attention to the looming stimuli. For hemianopic patients with unilateral lesions of the geniculostriate pathway, the ipsilesional SC and VTA showed significant activation to collision stimuli in their scotoma. Furthermore, stronger SC responses predicted better behavioral performance in collision detection even in the absence of awareness. Therefore, human tectofugal pathways could automatically detect collision trajectories without the observers' attention to and awareness of looming stimuli, supporting "blindsight" detection of impending visual threats.

## Introduction

Detecting objects approaching on a collision course is critical for the survival of animals in the environment. Specialized neurons or brain circuits highly sensitive to looming stimuli have been identified in many species, including insects [1], fish [2], pigeons [3], mice [4,5], and others. In primates, both adult monkeys and human infants elicit avoidance behaviors to symmetrically looming stimuli [6,7], imaging studies mainly revealed looming sensitive responses in the cortical brain regions [8–12]. These previous studies mostly compared a large looming stimulus versus translating or receding stimuli, whose motion directions greatly deviate from the collision course. The neural mechanism for computing the time-to-contact (TTC)

**Funding:** This study was supported by Ministry of Science and Technology of China (https://en.most. gov.cn/) STI2030-Major Projects (2022ZD0211900 to P.Z., 2021ZD0204200 to S.H.), National Natural Science Foundation of China (https://www.nsfc. gov.cn/english/site_1/index.html, 31871107 and 31930053 to P.Z., 82101110 to H.Z.), Chinese Academy of Sciences (https://english.cas.cn/, XDB32020200, KJZD-SW-L08, YSBR-071) to S.H., and Beijing Natural Science Foundation (https:// mis.kw.beijing.gov.cn/, 7212092) and the Capital's Funds for Health Improvement and Research (https://wjw.beijing.gov.cn/, 2022-2-5041) to D.W.. The funders had no role in study design, data collection and analysis, decision to publish, or preparation of the manuscript.

**Competing interests:** The authors have declared that no competing interests exist.

**Abbreviations:** AGFI, adjusted goodness of fit index; AttNet, attention network; BVF, blind visual field; CDF, cumulative distribution function; CFI, comparative fit index; DA, dopaminergic; FDR, false discovery rate; FEF, frontal eye field; FWE, family-wise error; FWHM, full-width half-maximum; GFI, goodness of fit index; GLM, general linear model; IFG, inferior frontal gyrus; INS, insular; ISI, interstimulus interval; LC, locus coeruleus; LGN, lateral geniculate nucleus; LME, linear mixed effect; LORO, leave-one-run-out; LOSO, leave-one-subject-out; LP, lateral posterior; MTG, middle temporal gyrus; NVF, normal visual field; PBGN, parabigeminal nucleus; PGFI, parsimony goodness of fit index; PV, parvalbumin; RMR, root mean square residual; RMSEA, root mean square error of approximation; SC, superior colliculus; SEM, structural equation modeling; TPJ, temporal parietal junction; TTC, time-to-contact; VC, visual cortex; vlPul, ventrolateral pulvinar; vmPul, ventromedial pulvinar; VTA, ventral tegmental area; 2-IFC, 2-interval forced choice.

information has been extensively studied and was suggested to provide warning detection for large approaching objects. However, humans are not only sensitive to looming versus non-looming stimuli but also efficient in detecting collision course from near-miss trajectories (i.e., a few centimeters away from the head at the pass point). For example, compared with a near-miss trajectory, an approaching object on a collision course with the observer could automatically capture visual attention, increase perceived object size and evoke greater pupil constrictions, even without conscious awareness of the motion direction [13,14]. These behavioral findings suggest early subcortical mechanisms for collision detection, through precise and sensitive measures of motion trajectories. However, little is known about the neural mechanism in the human brain for detection of collision trajectories.

The superior colliculus (SC) is a phylogenetically old visual nucleus lying on the roof of the mammalian brainstem. It plays important roles in visual perception and visually guided reorienting functions, such as attention, eye and head movements [15]. As a key retino-recipient region, the SC and its homologous structure in nonmammals, optic tectum, were found preferentially sensitive to looming stimuli in a number of species [4,5,8]. In rodents, specific types of neurons in the SC have been identified as the key components of several subcortical circuits to detect looming objects and trigger defensive responses, including the SC-PBGN (parabigeminal nucleus)-Amygdala [4,16], SC-LP (lateral posterior thalamic nucleus, homolog of primate pulvinar)-Amygdala [5,16] and SC-VTA (ventral tegmental area)-Amygdala [17], and LC (locus coeruleus)-SC [18] connections. Given that the subcortex is the primitive brain and that subcortical functions might be relatively conserved in mammals, the subcortical pathways found in the rodent brain might also play important roles for collision detection in the human brain. Alternatively, with the expansion of the neocortex and reduced retinal projection to the SC in higher mammals [15], it could be possible that the cerebral cortex plays a more prominent role in collision detection in the human brain. An early 3T fMRI study at a relatively low resolution suggests that the human SC, pulvinar, exhibited stronger responses to looming compared to receding stimuli [8]. However, it is unclear whether these subcortical responses were a result of cortical influence. Most importantly, the underlying neural circuitry for detecting collision trajectories remains unknown.

Moreover, it is unknown whether these collision detection mechanisms could work automatically even without the observer's attention to and awareness of the looming stimuli, as suggested by previous behavioral studies in humans [13,14]. Some patients with cortical blindness can detect or even discriminate visual stimuli presented to their blind visual field, despite denial of seeing the stimuli. This phenomenon, called blindsight, attracts broad interests and has been hotly debated for almost half a century [19]. Although lots of studies have been done, the neural pathways involved and their specific functional roles in blindsight remain highly controversial. Looming-evoked avoidance behavior was observed in monkeys with V1 lesions [20], suggesting a critical role of subcortical pathways in automatic "blindsight" detection of impending visual threats. However, direct evidence for this hypothesis is still lacking.

To answer these questions, the current study used high-resolution fMRI to investigate the neural pathways involved in detecting collision trajectory in both healthy human participants and hemianopic patients. The motion trajectory of an incoming object was varied slightly to be either on a collision course or on a near-miss trajectory regarding the head of observers. For healthy participants, we first measured their visual ability to discriminate collision course from near-miss trajectories, and the associated changes in pupillary reflex (experiment 1). In a 7T fMRI experiment (experiment 2), we further investigated the neural circuitries for detecting collision trajectories and examined whether these mechanisms require top-down attention or could operate automatically without paying attention to the stimuli. In a group of hemianopic patients with unilateral lesions of the geniculostrate pathway, experiment 3 studied whether

the tectofugal pathways are sensitive to collision trajectories even without awareness of stimuli presented in their blind visual field.

## Results

### Behavioral and pupil size sensitivity to collision trajectory

In a behavioral experiment (experiment 1, $n$ = 15), we measured the ability of healthy human observers to discriminate hit from near-miss trajectories. Participants responded whether an incoming object would hit or miss their head (Figs 1A and S1A). Pupil size and eye movements were also recorded. In Fig 1B, the percentage of hit responses was plotted as a function of the extrapolated impact point of the approaching objects. The psychometric functions were further fitted with cumulative normal distributions. The steep slope of the psychometric function around the edge of the head (about 6 cm from nasion) indicates that participants could precisely discriminate the incoming objects on a collision course from those with near-miss

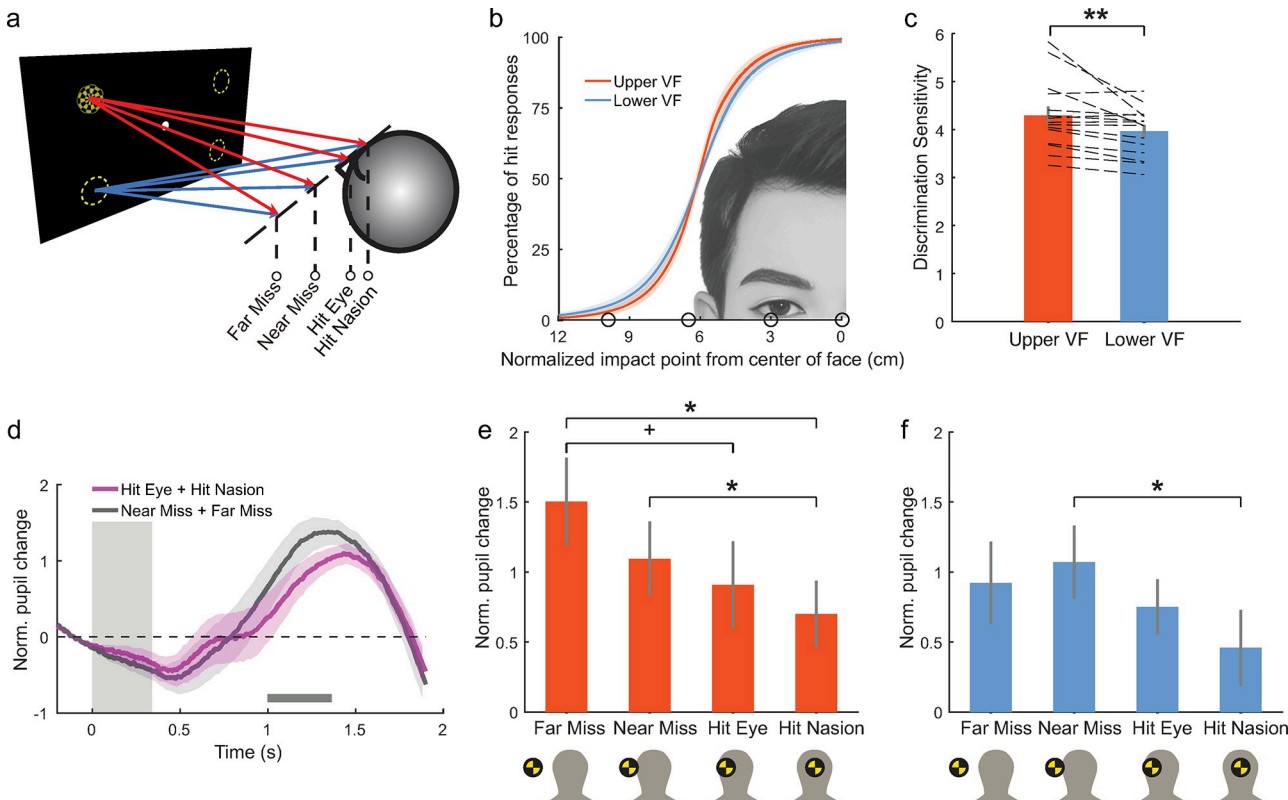

**Fig 1. Schematic stimulus diagram and results of the behavioral experiment (experiment 1). (a)** Visual stimuli depicting an incoming ball from one of the 4 quadrants of the VF were presented with a 3D LCD monitor in the behavioral experiment. The trajectory of the looming object varied slightly to either hit (hit nasion, hit eye) or miss (near miss, far miss) the head of observers. **(b)** The percentage of hit responses at different impact points was fitted with a normal CDF. Black circles indicate the extrapolated impact points of different trajectories. **(c)** The discrimination sensitivity, calculated as $\ln\left(\frac{1}{\sigma}\right)$ in which $\sigma$ is the standard deviation of the fitted normal CDF, is significantly higher in the upper VF than in the lower VF. Error bars indicate SE. **$p < 0.01$. **(d)** The change of pupil size with respect to baseline (−200~0 ms) was normalized by standard deviations and then plotted for hit (purple) and miss (dark gray) trajectories, respectively (data from the upper and lower VFs were combined here). Dark gray bars indicate the time points when the pupil size was significantly smaller (permutation test $p < 0.05$ using cluster-size based adjustment) in the hit condition. The vertical gray bar indicates the time interval of visual stimulus presentation. **(e, f)** The averaged change of pupil size from 1,000 to 1,364 ms after stimulus onset was plotted for different looming stimuli from the upper **(e)** and lower **(f)** VFs. *Bonferroni corrected $p < 0.05$, + uncorrected $p < 0.05$. Shaded areas in **(b, d)** and vertical lines in **(c, e, f)** indicate SEM. Data underlying **(c, e, f)** can be found at https://osf.io/gdjwh/. CDF, cumulative distribution function; SEM, standard error of the mean; VF, visual field.

trajectories, consistent with the prediction from previous studies [13,14]. S4A Fig shows the results for all individuals.

The discrimination sensitivity, calculated based on the slope of the psychometric curve, was slightly but significantly higher for objects approaching from the upper visual field than from the lower visual field (permutation test $p < 0.001$; Fig 1C), indicating that observers could more precisely discriminate hit from near-miss trajectories for objects coming from the upper visual field. Pupil size was smaller around 1,100 ms after stimulus onset in the hit than miss conditions (Fig 1D–1F), consistent with previous findings [13]. Similar results were observed when testing with a much brighter background or in an irrelevant task condition (S5 and S6 Figs). Fixational eye movements showed no significant difference between the hit and miss events (S4B Fig). These behavioral results show that observers and their pupillary responses can precisely discriminate collision from near-miss trajectories of an incoming object.

## Enhanced SC responses to looming objects on a collision course

To investigate potential subcortical pathways for the detection of collision trajectory in the human brain, we performed a 7T fMRI experiment (1.5-mm isotropic resolution) in a group of healthy human participants in experiment 2 ($n = 20$). In a long event-related design (Fig 2A), participants were asked to discriminate whether an incoming object would hit or miss their head in an attended condition or to detect occasionally and randomly presented fixation-color changes in an unattended condition. Participants' performance was 97.2% ± 3.6% (mean ± STD) in the attended condition and 95.6% ± 4.7% in the unattended condition, indicating that they followed the task instructions very well.

Previous studies suggest that the SC, a phylogenetically old midbrain nucleus, might play important roles for collision detection in the human brain [8]. Thus, in the current study, we carefully investigated the response of the SC to looming stimuli with direct-hit or near-miss trajectories. From the group-averaged activation maps (Fig 2C), bilateral SCs showed robust responses to the looming stimuli in both attended and unattended conditions. This confirmed that the human SC was indeed highly sensitive to looming stimuli. We further investigated the **collision sensitivity** in each voxel defined as the response difference between the direct-hit and near-miss trajectories (Fig 2C, "Hit-Miss"). In the attended condition, no significant cluster of voxels with collision sensitivity was found in the SC. However, significant or marginally significant clusters can be found in the unattended condition (permutation test with small volume correction: cluster $p = 0.053$ for the contralateral SC, and $p < 0.001$ for the ipsilateral SC. The cluster defining threshold is voxel $p = 0.05$). These collision-sensitive clusters located more rostral (or anterior) (Fig 2B, foveal SC, "Hit-Miss," unattended), corresponding to the foveal part of the SC [21,22]. More caudal (or posterior) part of the SC showed strong responses to both looming stimuli presented in the contralateral visual field (Fig 2C, extrafoveal SC, "Hit" and "Miss"), but no significant collision-sensitive clusters. Based on the normalized depth map of the SC (S5 Fig in our previous study [23]), we divided the SC into 3 compartments with equal thickness, roughly correspond to the superficial (depth from 0 to 1/3), intermediate (1/3 to 2/3), and deep layers (2/3 to 1). The center-of-mass location of the contralateral (depth = 0.2) and ipsilateral (depth = 0.36) clusters locate in the superficial and intermediate layers, respectively.

The ROI analysis focused on the responses to stimuli from the contralateral visual field. We first investigated the responses of the whole SC (Fig 2C). Two-way repeated measures (rm) ANOVA revealed a significant main effect of trajectory (Hit/Miss, $F(1,19) = 8.67$, $p = 0.008$, $\eta_p^2 = 0.313$) and attention (Attended/Unattended, $F(1,19) = 49.27$, $p < 0.001$, $\eta_p^2 = 0.722$). Post hoc paired $t$ tests showed a significant collision sensitivity in the unattended condition ($t(19) =$

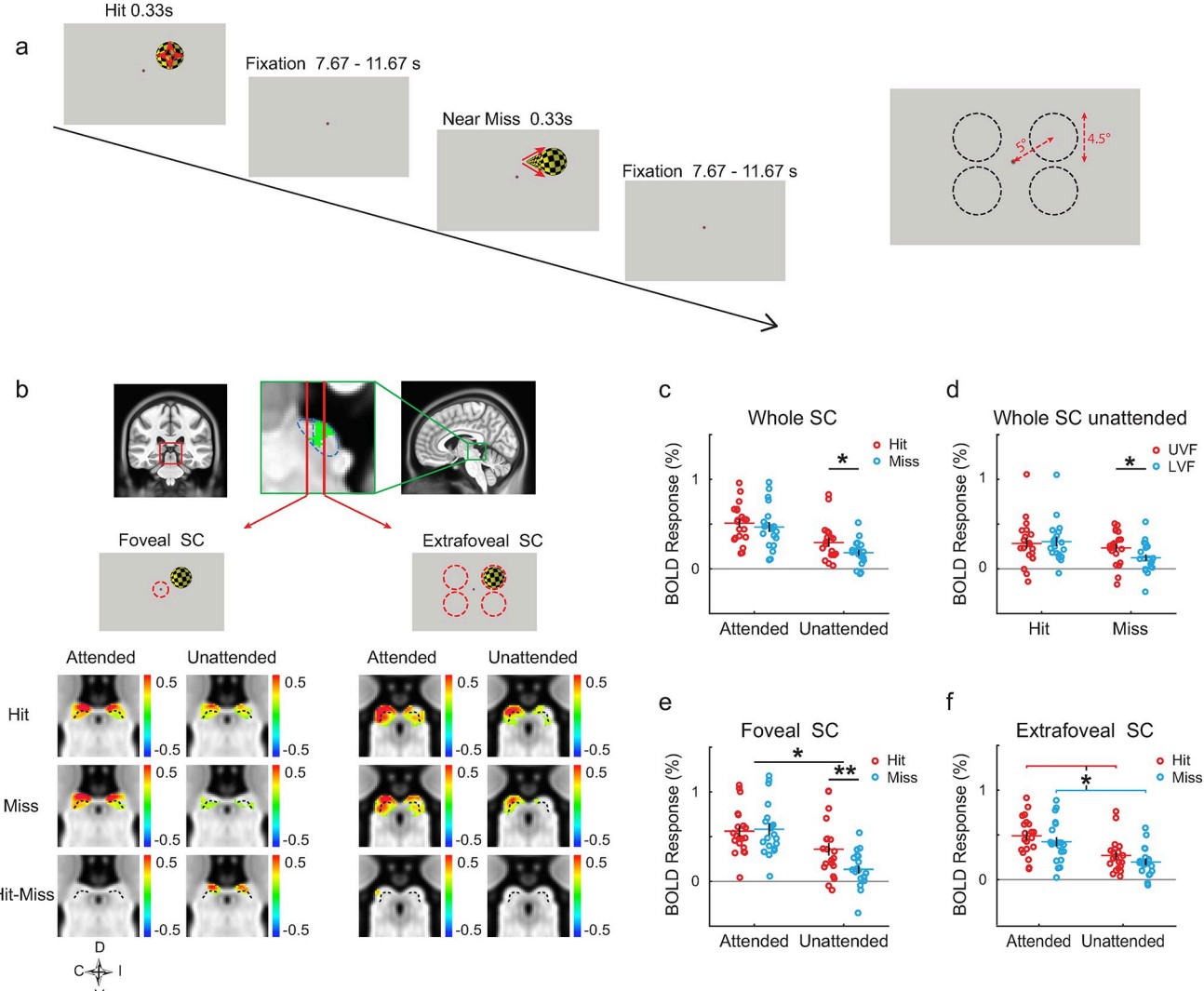

**Fig 2. Schematic diagram of stimuli and procedure and looming-evoked responses in the SC of healthy participants (experiment 2). (a)** Looming objects with hit and near-miss trajectories were presented in 2D images with an MRI-compatible projector for 330 ms, with 7.67~11.67 s of intertrial intervals. The stimulus was presented in one of the 4 quadrants of the VF in each trial. The right panel shows the size and position of the stimulus at the last frame in each quadrant. **(b)** SC activation maps. Red and green squares on the coronal and sagittal slices indicate the locations of the SC. From the zoomed-in sagittal view, blue dashed lines outline the ROIs of the rostral/anterior and caudal/posterior SCs representing the foveal and extrafoveal visual fields, respectively. The ROIs was determined by the retinotopic activations (contra-ipsi; S7 Fig) shown here as the green overlay. Activation maps were thresholded at $p < 0.05$ (uncorrected). Color bars indicate percent signal change. Left and right SCs were mapped with responses to the contralateral and ipsilateral stimuli, respectively. Activation maps to stimuli from the left visual field were horizontally flipped and averaged with those to stimuli from the right VF. C, I, D, and V in the compass abbreviate contralateral, ipsilateral, dorsal, and ventral, respectively. Dotted lines indicate an approximate boundary between the superficial and deeper layers of the SC. **(c and d)** ROI-averaged BOLD responses of the whole SC to looming stimuli from the contralateral visual field. **(e and f)** The responses in the foveal **(c)** and extrafoveal **(f)** parts of the SC. Red/blue horizontal bars, black vertical bars, and red/blue circles denote mean, SE, and individual data, respectively. * above a long black line indicates ANOVA $p < 0.05$ for the interaction between attention and trajectory. * and ** above a short black line indicate paired $t$ test $p < 0.05$ and 0.01, respectively. * In between the red and blue lines in (f) indicates $p < 0.05$ for the main effect of trajectory. The original fMRI resolution is 1.5-mm isotropic; here, the functional maps were up-sampled at 0.6-mm isotropic to match the resolution of MNI template (see Methods for details). Data underlying **(c, d, e, f)** can be found at https://osf.io/gdjwh/. SC, superior colliculus; SE, standard error; VF, visual field.

2.363, $p = 0.029$, Cohen's $d = 0.528$), but not in the attended condition ($p = 0.247$). We then investigated SC responses to looming stimuli in the upper and lower visual fields in the unattended condition (Fig 2D). Although collision sensitivity in the upper and lower visual fields showed no significant difference (trajectory by visual field interaction: $p = 0.124$), the SC

responses to looming stimuli on a near-miss trajectory were significantly stronger in the upper visual field compared to those in the lower visual field ($t(19) = 2.798$, $p = 0.011$, Cohen's $d = 0.626$), suggesting higher looming sensitivity in the SC to approaching objects from the upper visual field. This finding is consistent with the behavioral results of slightly better looming-trajectory discrimination and stronger pupillary reflex in the upper visual field (Figs 1, S5, and S6).

We further divided the SC into a rostral (foveal) part and a caudal (extrafoveal) part (enclosed by blue dotted lines in Fig 2B) based on the significant contralateral-ipsilateral activations (green voxels in the upper middle panel of Fig 2B; see also S7 Fig). In the foveal SC (Fig 2E), there was a significant interaction between trajectory and attention ($F(1,19) = 8.06$, $p = 0.010$, $\eta_p^2 = 0.298$). Post hoc $t$ tests showed significant collision sensitivity in the foveal SC in the unattended condition ($t(19) = 3.015$, $p = 0.007$, Cohen's $d = 0.674$), but not in the attended condition ($p = 0.600$). To further validate the collision sensitivity found in the foveal SC in the unattended condition, we performed a leave-one-subject-out (LOSO) cross-validation analysis. For each participant, the ROI to calculate the collision-sensitive response was defined by voxels with significant group-level collision sensitivity from the remaining participants. The LOSO results revealed a significant collision sensitivity in the foveal SC in the unattended condition ($t(19) = 2.761$, $p = 0.012$, Cohen's $d = 0.617$). In the extrafoveal SC (Fig 2F), there was a significant main effect of trajectory ($F(1,19) = 6.75$, $p = 0.018$, $\eta_p^2 = 0.262$), but no interaction between attention and trajectory ($p = 0.943$). This finding suggests collision sensitivity in the extrafoveal SC independent with the attentional state of observers.

To evaluate whether looming stimuli lead to any measurable head movement, we investigated head movements after the onset of the looming stimuli using the motion parameters estimated with EPI volumes. Results showed negligible head motion to looming stimuli (less than 0.01 mm on average). We further confirmed that the observed collision sensitivity in the SC was unlikely influenced by sparsely and randomly presented fixation changes or due to retinotopic difference in luminance changes and optical flows between hit and near-miss trajectories (S2 and S3 Figs, S1 Text).

## Collision sensitivities in the ventral pulvinar and VTA

We further investigated collision sensitivity in other subcortical regions, including the pulvinar, VTA, PBGN, amygdala, and LC, which form looming sensitive circuits with the SC as demonstrated by previous rodent studies [4,5,16–18]. The lateral geniculate nucleus (LGN) was also included as a potential control area. S8 Fig shows the anatomical masks for these subcortical regions in MNI space. For each ROI, we performed similar analyses as for the SC to check its activation map and ROI-averaged responses. Collision-sensitive responses were found in the pulvinar and VTA (Fig 3), but not in other subcortical nuclei (S9 Fig).

As shown by Fig 3A, looming stimuli activated both lateral and medial portions of the ventral pulvinar. The ventrolateral pulvinar (vlPul) mainly connects with the early visual cortex but also with the parietal cortex [24] and may also receive sparse input from the SC [25,26]. As shown by Fig 3A, a collision-sensitive cluster was found in the vlPul in the attended condition (cluster $p = 0.091$). In Fig 3B (left panel), the ROI-averaged responses of vlPul to stimuli presented to the contralateral visual field showed significant interaction of attention and trajectory ($F(1,19) = 4.62$, $p = 0.045$, $\eta_p^2 = 0.196$). Post hoc paired $t$ tests revealed a significant collision sensitivity in the attended condition ($t(19) = 2.750$, $p = 0.013$, Cohen's $d = 0.615$), but not in the unattended condition ($p = 0.347$). The LOSO cross-validation analysis revealed a significant collision sensitivity in the vlPul in the attended condition ($t(19) = 2.646$, $p = 0.016$, Cohen's $d = 0.592$).

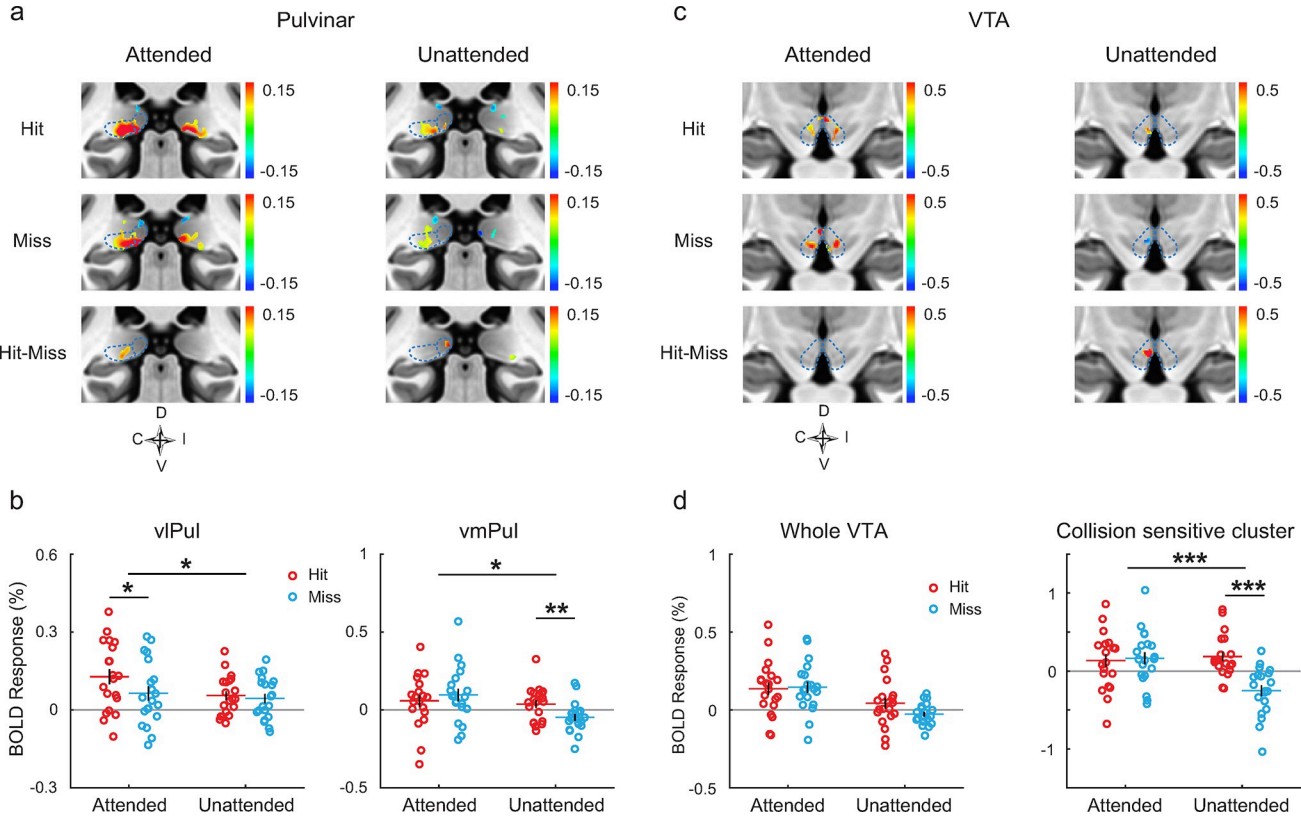

**Fig 3. BOLD responses to looming stimuli in the pulvinar and VTA (experiment 2). (a, c)** Activation maps to looming stimuli in coronal views of the pulvinar **(a)** and VTA **(c)**. Maps were thresholded at $p < 0.05$ uncorrected. Dotted lines mark the anatomical boundaries of the vlPul, vmPul, and VTA. Activation maps of the whole pulvinar were smoothed with a 2.8-mm FWHM Gaussian filter for display purpose (1.4 mm FWHM for other nuclei). **(b)** ROI-averaged responses of the vlPul and vmPul. **(d)** Left panel shows the ROI-averaged responses of the whole VTA. Right panel shows the averaged responses of the collision-sensitive cluster in **(c)**. * or *** above a long line indicate ANOVA $p < 0.05$ or $p < 0.001$. *, **, or *** above a short line for *t* test $p < 0.05$ or $p < 0.001$. Other conventions as in Fig 2. Data underlying **(b, d)** can be found at https://osf.io/gdjwh/. FWHM, full-width half-maximum; vlPul, ventrolateral pulvinar; vmPul, ventromedial pulvinar; VTA, ventral tegmental area.

The ventromedial pulvinar (vmPul) receives strong inputs from the SC and possibly sparse input from the retina and reciprocally connects with visual areas in the dorsal visual stream [26–28]. The dorsal part of vmPul may also connect with the amygdala and frontoparietal cortex [29,30]. In the group-averaged activation maps in the unattended condition (Fig 3A), there was a small cluster of collision sensitivity (Fig 3A, bottom right). The ROI-averaged responses of vmPul (Fig 3B, right panel) showed a significant interaction between trajectory and attention ($F(1,19) = 7.06$, $p = 0.016$, $\eta_p^2 = 0.271$). Post hoc *t* tests revealed a significant collision sensitivity in the unattended condition ($t(19) = 3.613$, $p = 0.002$, Cohen's $d = 0.808$), but not in the attended condition ($p = 0.321$). LOSO cross-participants validation did not find significant effect of collision sensitivity, likely due to a relatively large interindividual variability of collision-sensitive response in the vmPul. No significant collision-sensitive response was found in other subnuclei of pulvinar.

The VTA receives direct input from the SC [17,31,32] and projects to many brain areas including the amygdala [17] and frontal lobe [33]. In the attended condition (Fig 3C, left), the VTA showed significant activations to looming stimuli but without collision sensitivity. In the unattended condition, there was a significant collision-sensitive cluster (cluster $p = 0.004$). The averaged responses of the collision-sensitive cluster were plotted in the right panel of Fig 3D. For the ROI-averaged response of the whole VTA (Fig 3D, left panel), we found a marginally

significant effect of collision sensitivity in the unattended condition ($t(19) = 1.970$, Cohen's $d = 0.441$, $p = 0.064$). LOSO analysis further revealed a significant collision sensitivity in the unattended condition ($t(19) = 3.234$, Cohen's $d = 0.723$, $p = 0.004$).

As multiple tests of collision sensitivity were performed in several subcortical ROIs, we further controlled family-wise errors with a sequential Holm–Bonferroni approach (see statistical analysis in Methods for details). After correction, the foveal SC, vmPul, and VTA still shows significant collision sensitivity in the unattended condition (all $p < 0.05$).

## Collision sensitivities in the visual and frontoparietal cortices

We further investigated whether collision sensitivity can also be found in the cortical brain regions. ROI-averaged retinotopic responses in a visual cortical area were determined by a leave-one-run-out (LORO) cross-validation approach at the individual level (see Methods for details). As shown by Fig 4A, significant collision sensitivity can be found in the early visual cortex in both attended and unattended conditions (attended condition: false discovery rate (FDR) $p = 0.030$ in V3, $p = 0.030$ in V3b, and $p = 0.030$ in V4 after BH-FDR correction; uncorrected $p = 0.027$ in V1; unattended condition: uncorrected $p = 0.044$ in V4, uncorrected $p = 0.031$ in TO2). Fig 4B shows the group-averaged collision-sensitive (Hit-Miss) activations on the cortical surface. A few clusters with collision sensitivity ($p < 0.01$, uncorrected) can be found from bilateral middle temporal gyrus (MTG) in the attended condition and from bilateral temporal parietal junctions (TPJ), and the frontal eye field (FEF), inferior frontal gyrus (IFG), and insular (INS) in the right hemisphere in the unattended condition. However, no significant cluster can be found after family-wise error correction. These results revealed collision-sensitive responses in the visual cortex and possibly in the frontoparietal areas of attention network.

## SC-vmPul and SC-VTA pathways detect collision without attention

To further investigate the subcortical circuits involved in collision detection and the potential influence from cortical brain areas, we calculated across-participant correlations between

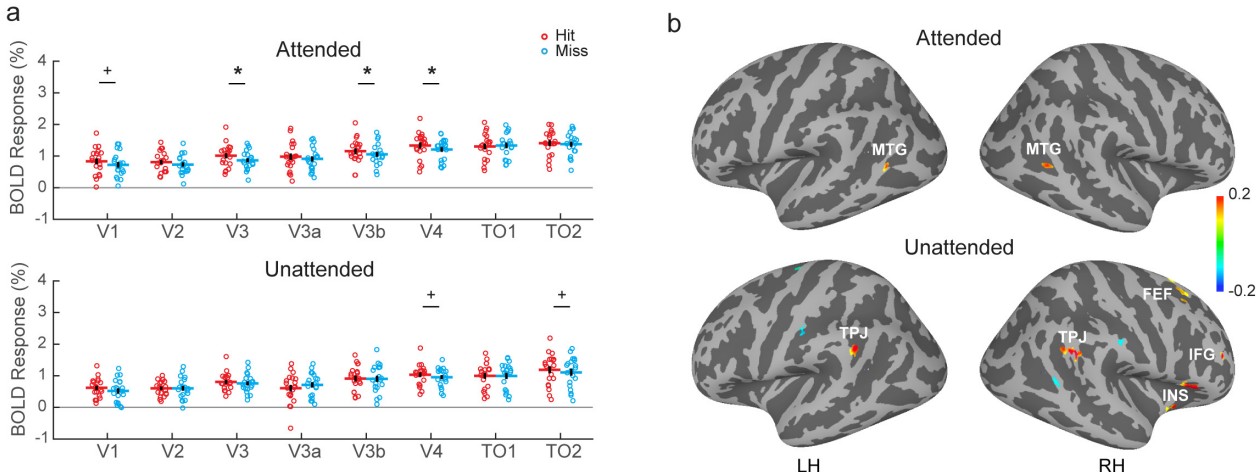

**Fig 4. Collision sensitivity in the cortical areas (experiment 2). (a)** BOLD responses in the visual cortex. *, **, or *** denote $p < 0.05$, $p < 0.01$, or $p < 0.001$ after FDR correction, and + for $p < 0.05$ (uncorrected). **(b)** Collision-sensitive activations in the high order visual cortex and the frontoparietal areas. The color bar indicates the percent signal change of Hit-Miss. Maps were thresholded at uncorrected $p < 0.01$. No significant clusters can be found after FWE correction. Data underlying **(a)** can be found at https://osf.io/gdjwh/. FEF, frontal eye field; FWE, family-wise error; IFG, inferior frontal gyrus; INS, insular; LH, left hemisphere; MTG, middle temporal gyrus; RH, right hemisphere; TO, temporal occipital; TPJ, temporal parietal junction.

collision-sensitive responses in the subcortical nuclei (SC, vmPul, and VTA) and cortical areas including the visual cortex (VC) and frontoparietal attention network (AttNet), followed by a path analysis with structural equation modeling (SEM) on the beta series of looming-evoked responses to infer their effective connectivity. We selected these ROIs because significant collision-sensitive responses were found in these regions. The vlPul was not included in this analysis because the SC's projection to this area is weak and controversial [25,26], and the collision-sensitive responses in the SC and vlPul showed no significant correlation (uncorrected $p > 0.4$ in both attended and unattended conditions).

In the attended condition ([Fig 5A], left panel), we found significant correlations of collision-sensitive responses between the SC and its downstream subcortical target vmPul ($r = 0.556$, uncorrected $p = 0.014$), between VC and VTA ($r = 0.478$, uncorrected $p = 0.033$), and between VC and AttNet ($r = 0.651$, $p = 0.021$, after family-wise error (FWE) correction of the correlation matrix). In the unattended condition ([Fig 5A], right panel, and 5b), there were significant correlations between the SC and both downstream subcortical targets (SC-vmPul: $r = 0.593$,

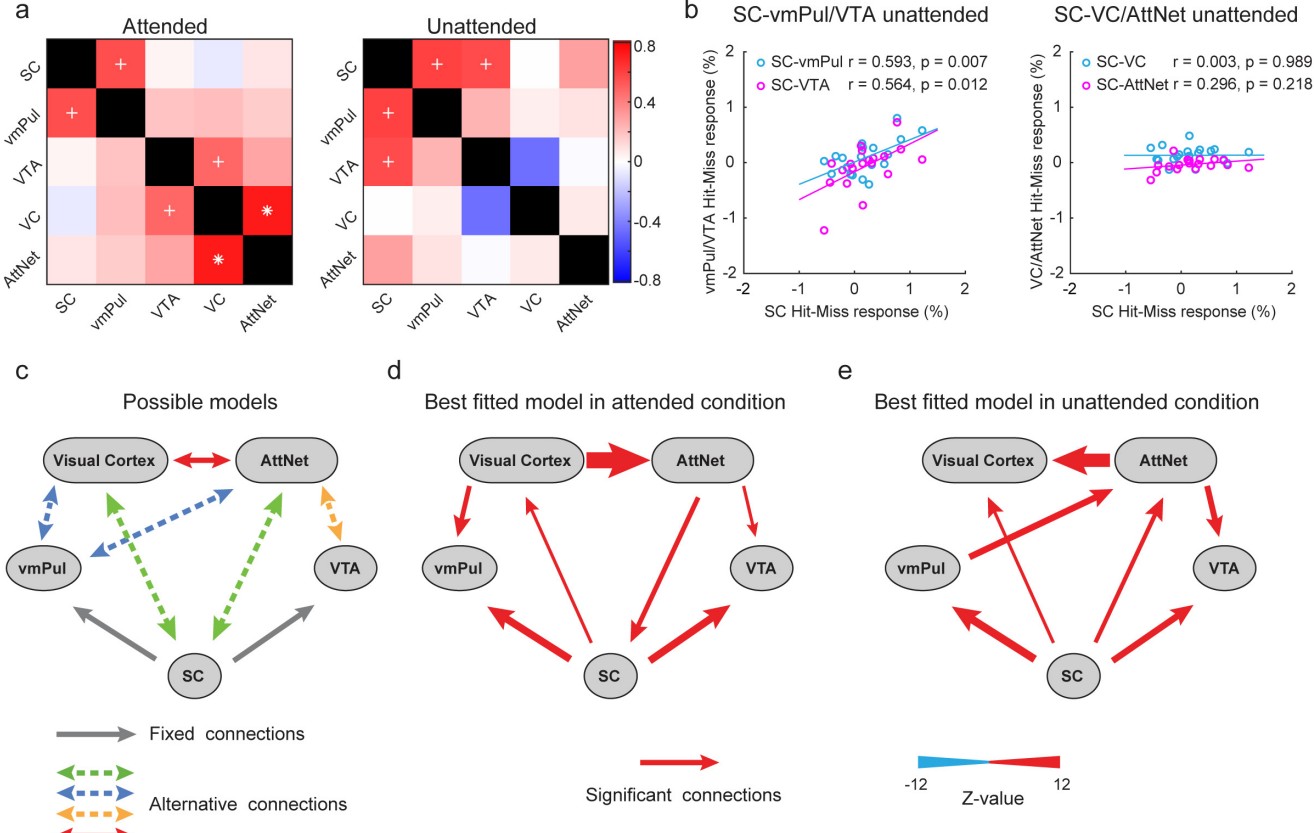

**Fig 5. Correlation of collision sensitivity and path analysis of beta series between cortical and subcortical regions (experiment 2). (a)** Correlation matrix of collision sensitivity. Each grid in the matrix shows the Pearson's correlation coefficient between the collision sensitivities of 2 ROIs, in the attended (left) and unattended (right) conditions. + indicates uncorrected $p < 0.05$ and * for $p < 0.05$ after FWE correction by permutation test. Color bars indicate the size of correlation coefficient. **(b)** Correlations between the collision sensitivity in the SC, and those in subcortical (vmPul/VTA) and cortical (VC/AttNet) areas. Each dot represents 1 participant. **(c)** Candidate SEM models for the effective connectivity of beta series between ROIs. Gray solid single-headed arrows represent fixed one-way connection. Red solid double-headed arrows indicate connections with 2 alternative directions. Dotted double-headed arrows indicate that the connection may follow one of 2 alternative directions or may not exist. All combinations of alternative connections yield 216 candidate models. **(d, e)** Best fitted models in the attended **(d)** and unattended **(e)** conditions. Arrows with solid lines indicate significant connections. There was no insignificant connection. Data underlying **(b)** can be found at https://osf.io/gdjwh/. AttNet, frontoparietal attention network; FWE, family-wise error; SC, superior colliculus; VC, visual cortex; vmPul, ventromedial pulvinar; VTA, ventral tegmental area.

uncorrected $p = 0.007$; SC-VTA: r = 0.564, uncorrected $p = 0.012$), but not between cortical and subcortical regions (all uncorrected $p > 0.1$).

Using a beta-series analysis [34] of the looming-evoked responses, we found significant functional connectivity between SC and vmPul and between SC and VTA in both attended and unattended conditions (all $p < 0.001$, Holm corrected). To identify the subcortical pathways from the SC and the information flow between cortical and subcortical areas, we performed a SEM path analysis with the beta series from these ROIs. A total of 216 candidate models were constructed with different combinations of 6 sets of alternative connections (Fig 5C), based on the known anatomical connections between these areas: SC-cortex [15], SC-vmPul [35], SC-VTA [17], vmPul-cortex [24], VTA-cortex [33]. Candidate models were fitted with the observed data and compared by the goodness of fit (see Methods for model comparisons). The best-fitted models for both attended (fit index: $\chi^2 = 18.481$, df = 2, CFI = 0.975, GFI = 0.997, AGFI = 0.978, PGFI = 0.133, RMSEA = 0.057, RMR = 0.019) unattended (fit index: $\chi^2 = 21.370$, df = 2, CFI = 0.968, GFI = 0.997, AGFI = 0.975, PGFI = 0.133, RMSEA = 0.062, RMR = 0.022) conditions revealed highly significant SC-vmPul and SC-VTA connections (all $p < 0.001$). Importantly, the model shows no cortical influence to the SC in the unattended condition. These findings support a pivotal role of the SC-vmPul and SC-VTA pathways in imminent collision detection even without attention. The collision sensitivity found in the SC was unlikely a result of cortical influence.

## Collision sensitivity in the ipsilesional SC of hemianopic patients

To further investigate whether the tectofugal pathways can detect collision trajectories even without awareness of visual stimuli and a functional geniculostriate pathway, we scanned a group of homonymous hemianopic patients with unilateral lesions of the geniculostriate pathway (experiment 3, $n = 12$). Patients lost their conscious vision of both eyes in one side of the visual field (see S1 Table for clinical characteristics and S11 Fig for visual field loss and lesioned locations). During fMRI scans, patients performed a central fixation task while stimuli with hit, near-miss, or receding trajectories were presented to their normal visual field (NVF) or blind visual field (BVF). Four patients (P09 to P12) also participated in a behavioral visibility test, in which they reported clear perception of stimuli presented to their NVF but denied seeing stimuli from the BVF. For most patients (8 of 12), there was no significant V1 activation to stimuli presented in their BVF (S11 Fig). Four patients showed weak uncorrected activations in their lesioned hemisphere, but no collision sensitivity.

Based on the findings of healthy participants, we focused our analysis on the SC, vmPul, and VTA in hemianopic patients. From the group-averaged activation maps in Fig 6A, significant clusters of activation to hit trajectories were found in the contralateral SCs regardless of whether the stimuli were presented in the NVF (cluster $p = 0.023$) or in the BVF (cluster $p = 0.031$). In the contralateral vmPul (Fig 6B) and VTA (Fig 6C), collision-sensitive clusters to stimuli presented to the BVF (voxel $p < 0.05$, uncorrected) were also found in similar locations as the results of heathy participants (Fig 3). To increase statistical power given the small number of hemianopic patients, we performed a linear mixed effect (LME) analysis with ROI-averaged responses of visually responsive voxels significantly activated by the receding stimuli (group-level $p < 0.05$ uncorrected). In the LME model, trajectory (Hit/Miss), visual field (NVF/BVF), and ROI (contralateral SC/vmPul/VTA) were defined as the fixed effects and participants as the random effect. The analysis revealed a significant hit response averaged across all 3 ROIs in the BVF (Fig 6E, $z = 2.59$, $p = 0.038$, Holm corrected across visual fields and trajectories), post hoc tests revealed significant hit responses in the SC ($z = 2.054$, $p = 0.04$ uncorrected) and VTA ($z = 2.768$, $p = 0.017$, Holm corrected across ROIs). A significant hit response

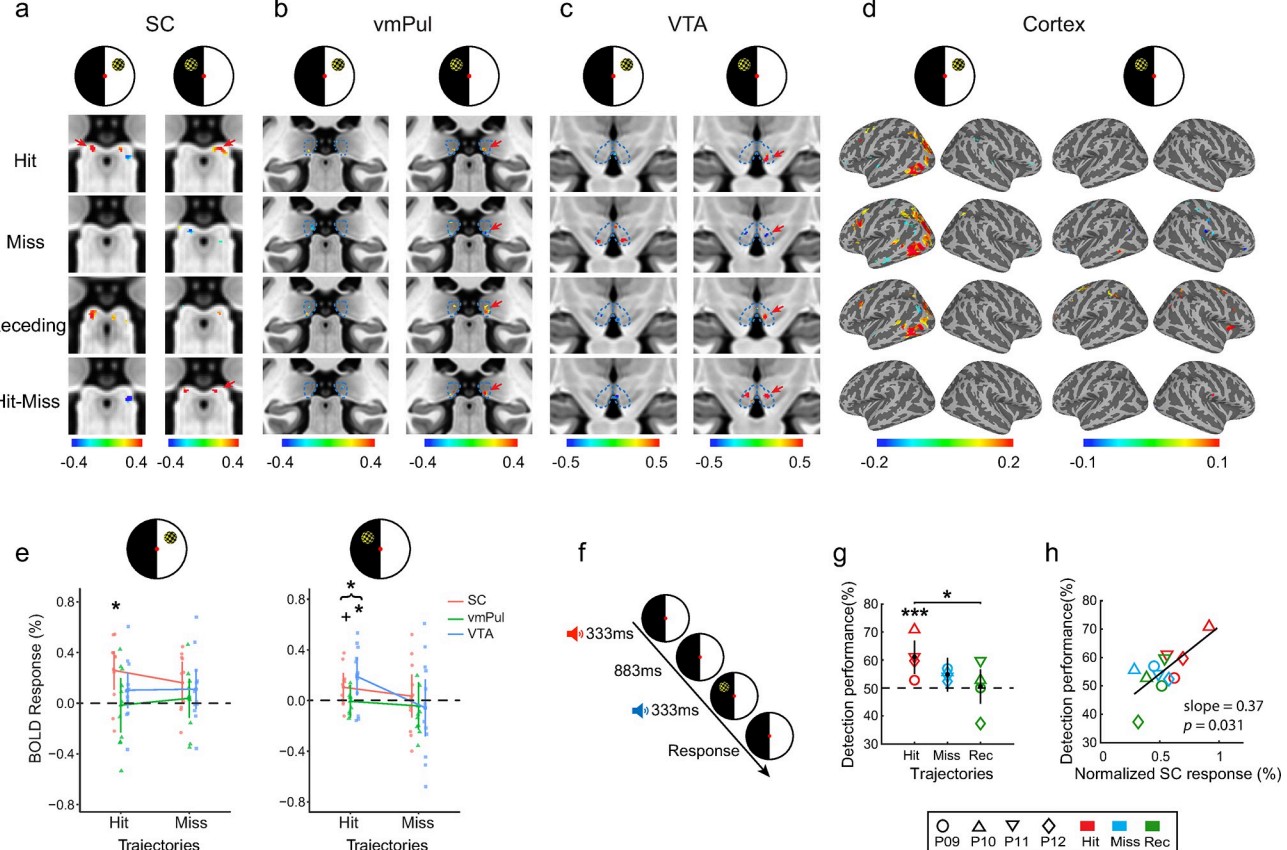

**Fig 6. fMRI results and behavioral performance of hemianopic patients (experiment 3). (a)** Group-averaged activation maps in the SC to stimuli presented to the NVF (left column) and BVF (right column). Maps were thresholded at $p < 0.05$ uncorrected. Color bars represent percent BOLD signal change. The left and right SCs were mapped contralateral to the NVF and BVF, respectively. **(b, c)** Activation maps in the vmPul **(b)** and VTA **(c)**. Conventions are the same as in **(a)**. Only activations within the ROI were shown. Red arrows indicate the location of collision-sensitive clusters from the Hit-Miss map. **(d)** Stimulus activations mapped to the cortical surface (thresholded at $p < 0.01$ uncorrected). Left columns show contra- and ipsilateral responses to stimuli presented to the NVF. Right columns show ipsi- and contralateral responses to stimuli presented to the BVF. **(e)** BOLD responses in the contralateral SC, vmPul, and VTA to looming stimuli presented to the NVF (left panel) and BVF (right panel). Each dot represents data from 1 patient. + and * represent $p < 0.05$ before and after Holm corrections, respectively. Error bars represent the 95% confidence intervals estimated in a LME model. **(f)** Schematic diagram and procedure for the 2-IFC detection task. **(g)** The detection permanence fitted with a binomial generalized linear mixed model. Error bars represent the 95% confidence intervals. * Post hoc test $p < 0.05$ for comparing marginal means between conditions. *** $p < 0.001$ for comparing marginal means to the chance level (50%). **(h)** The detection performance to stimuli in the BVF showed a linear relationship with the response of ipsilesional SC (in 10% most responsive voxels, chosen separately for each stimulus type). The black solid line denotes the fixed effect (SC response) of the linear mixed model. For each participant, SC responses were normalized by dividing the mean across 3 stimulus conditions and then multiplying the mean across all 4 participants. Data underlying **(e, g, h)** can be found at https://osf.io/gdjwh/. BVF, blind visual field; LME, linear mixed effect; NVF, normal visual field; Rec, receding stimulus; SC, superior colliculus; vmPul, ventromedial pulvinar; VTA, ventral tegmental area; 2-IFC, 2-interval forced choice.

was found in the contralateral SC to stimuli presented in the NVF ($z = 3.468$, $p = 0.012$, Holm corrected across all 12 conditions).

The LGN was also investigated since previous studies suggest that it plays a critical role in blindsight [36]. However, no collision-sensitive response was found in the LGN (S12 Fig). In the cortical regions, V1 (S11 Fig) and frontoparietal areas (Fig 6D) showed strong activations to stimuli presented in the NVF but not in the BVF. These findings demonstrate that the tecto-fugal pathways can automatically detect collision trajectories even without awareness of looming stimuli and a functional geniculostriate pathway.

## SC responses predict "blindsight" detection of collision

To check potential "blindsight" to approaching objects on a collision course, 4 patients (P09 to P12) also performed a 2-interval forced choice (2-IFC) detection task. The stimulus was presented in one of two 330-ms intervals with a 330-ms gap in between, accompanied by a low-pitch tone and a high-pitch tone in the first and the second intervals, respectively (Fig 6F). Patients had to determine in which interval the stimulus was presented. The detection performance was fitted with a binomial generalized linear mixed model with a random effect of participant (Fig 6G). Results showed a significant main effect of trajectory ($\chi^2(2)$ = 6.418, $p$ = 0.04). Following tests showed that the accuracy for the object on a collision course was significantly higher than that on a receding trajectory (contrast = 10.6%, $z$ = 2.529, $p$ = 0.023 using Holm adjustment) and then 50% chance level (estimated marginal mean = 61.1%, $z$ = 3.541, $p$ < 0.001). Importantly, the detection performance showed a significant linear relationship with the response of ipsilesional SC (main effect of SC response on detection performance in a linear mixed model with participant as the random effect: $F(1, 3.05)$ = 14.616, $p$ = 0.031, Fig 6H). No significant linear relationship was found between behavioral performance and the responses of vmPul or VTA. Altogether, the behavioral and fMRI results of hemianopic patients provide strong evidence in humans that the tectofugal pathways support blindsight to impending collisions.

## Discussion

Detecting imminent collision is crucial for our survival. In experiment 1, we found that human observers can precisely discriminate whether an approaching object was on a collision course or a near-miss trajectory with their head. Collision events also induced significant changes in pupillary reflex. In experiment 2, high-resolution 7T fMRI revealed collision-sensitive responses in several subcortical nuclei, including the SC, ventral pulvinar and VTA. Correlation and path analyses further demonstrated collision sensitivities in the SC-vmPul, and SC-VTA pathways without attention and cortical influence. In experiment 3, for hemianopic patients with unilateral lesions of the geniculostriate pathway, the ipsilesional SC showed collision sensitivity to stimuli presented to their BVF. Finally, stronger response in the SC was associated with better detection performance of the collision event. These findings clearly demonstrate a critical role of the human tectofugal pathways in automatic detection of collision trajectories without attention and awareness, supporting "blindsight" to threating visual information.

In the optic tectum and the downstream nucleus rotundus (homologues of the SC and pulvinar in mammals) of pigeons, different types of neurons encode several optical variables of a looming stimulus, including the time-to-collision, absolute rate of expansion, and object size [3]. In the mouse SC, Shang and colleagues identified parvalbumin-positive (PV+) excitatory projection neurons in the superficial layers encoding the optical parameters of a looming stimulus in their receptive fields [4,16]. The response onset depended on the stimulus size and moving velocity, and the response peaked at the time of collision. Here, we show that the human SC was highly sensitive to slight trajectory differences of looming stimuli with similar times to contact, rates of expansion, and object sizes, suggesting that the primate SC contains neurons sharply tuned to looming trajectories for accurate collision detection. Interestingly, although the looming stimuli were presented in the parafovea, the strongest collision-sensitive response was observed in the anterior part of the SCs corresponding to the central visual field (Fig 2C). One possibility is that SC neurons are sensitive to the would-be point of collision, making predictions about the impact point in the immediate future. Consistent with this explanation, predictive remapping activity has been observed for SC neurons before saccades

[37]. Collision-sensitive activations in the foveal SCs cannot be explained by retinotopic difference in optical flow or luminance, because the hit stimulus shows no overall retinotopic bias in the central visual field and the collision-sensitive clusters is clearly located outside the retinotopic region (S2B Fig). It was also unlikely due to fixational eye movements, because observers can maintain stable fixations that showed no significant difference in fixational eye movements between the hit and miss conditions (S4B). The lack of collision sensitivity in the attended condition was likely due to the saturation of SC responses.

Our results consistently show that the behavioral performance and pupillary reflex can better discriminate looming trajectories from the upper visual field (Figs 1, S5, and S6). The SC response was also stronger to looming (near-miss) stimuli in the upper visual field (Fig 2D). This upper visual field advantage of looming sensitivity suggests that the phylogenetically conserved tectal pathways are also ecologically adaptive, as threatening looming objects such as diving predators or falling stones appear more often in the upper than in the lower visual field due to gravity. A recent study showed that the primate SC, like that of rodents, also overrepresents the upper visual field, with sharper, stronger, and faster visual representations than in the lower visual field [22]. Together, these findings support Previc's ecological perspective of the primate visual system, suggesting that the primate SC may be optimized and specialized for detection of transient and biological salient information from extrapersonal space in the upper visual field [38]. This upper visual field advantage forms an interesting contrast with previous findings that sustained visual attention resolves and tracks objects better in the lower visual field [39].

The results of experiment 2 clearly show that compared with the looming-evoked responses during a central fixation task, paying attention to and judging the trajectory of looming stimuli strongly enhanced the responses in both superficial and deeper layers of the SC (Fig 2B). However, in experiment 3, the SC's responses to collision trajectories were comparable when the stimulus was presented to the NVF or to the BVF of hemianopic patients (Fig 6A). Thus, looming-sensitive responses in the SC were strongly modulated by top-down attention but may not be sensitive to the awareness of visual stimuli. These findings demonstrate the critical role of the SC in both top-down attention and subconscious processing of threatening visual information.

Our data clearly demonstrate that the SC-vmPul pathway in the human brain automatically detects collision trajectories without attention to the looming stimuli (Fig 5). Results from the hemianopic patients further suggest collision sensitivity in the ipsilesional SC and vmPul in the absence of visual awareness (Fig 6). These findings are consistent with recent rodent studies that the SC-LP (or pulvinar) pathway processes looming information in an anesthetized state [16] and triggers defensive freezing behavior [5,16]. In Wei and colleagues' study, optogenetic mapping and electrophysiology also revealed a disynaptic circuit from the SC through LP to the lateral amygdala, which directly mediated the innate fear-related defensive response. In our study, although looming objects on a collision course changed pupillary reflex, participants did not report fear to these stimuli, and no significant response was found in the amygdala (S10 Fig). One possible explanation is that human observers quickly adapted to repetitive presentations of the virtual looming objects, which may not be effective to trigger fear response or evoke strong activation in the amygdala. In contrast, collision-sensitive responses can be reliably observed in the SC and vmPul. We speculate that the role of the SC-vmPul pathway might be automatically processing collision-related visual information, such as the looming trajectory or time-to-collision, which could be used by the downstream brain areas (e.g., the dorsal stream) to guide quick and subconscious actions to avoid the impending threats [40].

Collision-sensitive activity was also found in the VTA without attention (Fig 3C and 3D) and awareness (Fig 6C and 6E), showing correlation with and effective connectivity from the

SC (Fig 5B and 5E). These findings are consistent with the rodent studies that VTA neurons responded in short latency to biologically salient or aversive stimuli [31] and that VTA neurons receiving direct input from the SC mediated defensive flight behaviors to large overhead looming stimuli [17]. Populated mainly with dopaminergic (DA) neurons, the VTA plays important roles in reward, motivation, and attention [41]. The role of SC-VTA pathway might be modulating the level of arousal [42] or changing the attentional state [43] of the observer to mediate rapid defensive responses [44]. Although rodent studies also suggest the PBGN and LC in processing looming information and mediating looming-evoked defensive behaviors [4,18], we did not find significant collision-sensitive response from these areas in the human brain. Given their small sizes and deep locations, the negative finding from these small subcortical nuclei could be due to the low SNR and partial volume effect of BOLD signals.

The tectopulvinar pathway was originally proposed to support "blindsight" [19]. A critical role of the SC in visuomotor functions has been repeatedly confirmed by studies of visually guided saccades in V1-lesioned monkeys [45,46]. FMRI studies of blindsight patients also indicated visually evoked activation from the SC [47,48]. However, given the small sample sizes and the low-resolution fMRI approaches in these studies, the involvement of the SC in human blindsight still lacks conclusive evidence. Using high-resolution fMRI, here we showed clear evidence from 12 hemianopic patients that visually evoked response in the ipsilesional SC was highly sensitive to looming objects on a collision course (Fig 6A and 6E), which also predicted the above-chance detection performance of impending collisions (Fig 6G and 6H). These findings provide strong evidence for the critical role of the SC in detecting impending visual threats in human blindsight.

Compared with the SC, the role of pulvinar in blindsight is more controversial. Human fMRI studies suggest that the tectopulvinar-amygdala pathway may underlie blindsight of emotionally and socially salient face stimuli [49,50]. A recent study in V1-lesioned monkeys also revealed a critical role of the tectopulvinar pathway in visually guided saccades [51]. However, there is accumulating evidence that the geniculo-extrastriate pathway plays a major role in blindsight in both Monkey [36] and Human [52]. These discrepancies might depend on the scope of lesion, time of lesion, or distinct roles of the pulvinar and LGN pathways in blindsight [53,54]. Our results may suggest collision sensitivity in the ipsilesional ventromedial pulvinar to looming stimuli presented to the BVF (Fig 6B). Collision-sensitive response was also found in the VTA (Fig 6C and 6E). While visually evoked responses can be found in the LGNs of both healthy participants (S10 Fig) and hemianopic patients (S12 Fig), there was no significant activation of collision sensitivity. These findings support distinct roles of the subcortical pathways in blindsight: The tectofugal pathways are specialized to detect impending visual threats, while the geniculo-extrastriate pathway processes basic visual information (e.g., contrast and motion).

Human fMRI studies reported cortical representations of the TTC and possibly trajectories of approaching objects [55]. While the cortical responses may indicate a complex and relatively slow neural calculation for 3D motion perception, the subcortical pathway found in our study suggests an efficient and rapid computational strategy for collision trajectory detection. In support of this, studies have demonstrated that simpler equations for calculating looming trajectories could better predict human performance, especially the systematic errors made by observers. For example, Duke and Rushton [56] suggested that the perceived trajectory is based on the ratio of lateral angular speed to the sum of looming and changing disparity signals. Our findings revealed the neural substrate for an efficient and rapid neural computation of collision trajectory in the human brain, which may inspire the optimization of computer vision algorithms for collision detection [57].

## Methods

### Experiment 1: Behavior and eye tracking in healthy participants

**Participants.**   A total of 15 healthy participants (20 to 30 years old, mean age = 24.6 years, SD = 2.16 years, 10 females) participated in experiment 1. They had normal or corrected-to-normal vision without neurological or psychiatric disorders. All participants (including those for experiments 2 and 3) gave written informed consent in accordance with procedures and protocols approved by the Institutional Review Board of the Institute of Biophysics, Chinese Academy of Sciences (2012-IRB-011).

**Stimuli and procedures.**   Visual stimuli were generated with Psychtoolbox 3.0 [58] in MATLAB (2017a). 3D-rendered spheres depicted from slightly different perspectives were projected into 2 eyes to produce a 3D effect. Stimuli were stereoscopically presented with shutter glasses (NVIDIA 3D VISION) and a compatible LCD display with 120 Hz refresh rate (60 Hz for each eye). Eye positions and pupil size of the left eye were recorded at 1,000 Hz with an Eyelink1000Plus system. A standard 9-point calibration was performed at the beginning of each session. A low-pass filter with cutoff frequency of 40 Hz was performed to the eye-tracking data to reduce the 60 Hz artifacts from shutter glasses. After interpolating the missing data due to eye blinks, pupil size and eye position time series were epoched and averaged across trials per condition and participant.

As shown in S1A Fig, the 3D display presented a baseball-sized sphere in 6-cm diameter moving at a speed of 24 m/s from 11.3 meters to 3.3 meters in front of the observer. The trajectory started from one of the 4 quadrants at a 0.65-meter horizontal offset and a 0.38-meter vertical offset and ended in the same quadrant. The would-be point of collision varied from the middle of the face (0 cm, hit nasion) to 3 cm (hit eye), 6 cm (near miss), or 12 cm (far miss) of horizontal offset. To match their retinotopic locations, the on-screen projections of looming trajectories were aligned based on the center of mass of projected images. The on-screen display was a sphere with a texture of black-and-yellow checkerboard expanding from 0.3 to 1 degree of visual angle in 330 ms, at an eccentricity of 3.8 degrees (angle with vertical meridian is 60˚; see S1 Fig for more details on screen size, starting and ending eccentricity, etc., for all 3 experiments). Participants were instructed to keep fixation and press buttons to report whether the looming object would hit or miss their head. For each participant, 400 trials were collected.

### Experiment 2: fMRI study with healthy participants

**Participants.**   A total of 20 healthy participants (22 to 42 years old, mean age = 26.3 years, SD = 4.0 years, 11 females) with normal or corrected-to-normal vision and no neurological or psychiatric conditions participated in experiment 2.

**Stimuli and procedures.**   Visual stimuli were rendered in 3D but presented with an MRI-safe 2D projector on a translucent screen behind the head coil. Participants viewed the stimuli through a mirror mounted inside the head coil. Looming objects (S1B Fig) were simulated to approach from 8.75 m from the observer and vanish at 0.75 m, at a speed of 24 m/s. The ball would either hit the eye of the observer to cause a collision (hit) or slightly miss the head (near miss) with a 6-cm horizontal offset. Same as in experiment 1, the on-screen projections of the 2 trajectories in the same quadrant were aligned based on the center of mass of projected images. The luminance change and overall magnitude of optical flows (S2 Fig) were similar between the 2 trajectory conditions. The stimulus on the screen expanded from 0.4 to 4.5 degrees of visual angle in 330 ms, at an eccentricity of about 5 degrees (angle with vertical meridian is 60˚).

In the attended condition, participants were instructed to keep fixation and respond whether the incoming object was on or off a collision course with their heads. In the unattended condition, they were instructed to keep fixation and detect occasional color changes of the fixation point. Participants performed 4 runs each for the attended and unattended conditions. The attended and unattended runs were scanned in alternation. Each run comprised 32 trials with an interstimulus interval (ISI) randomly chosen from 8, 10, and 12 seconds. Thus, 16 trials were collected for each combination of attention, trajectory, and quadrant conditions. Participant S19 lost one trial of data in some conditions due to a technical problem.

**MRI data acquisition.** MRI data were acquired with a 7T scanner (Siemens MAGNE-TOM, Erlangen, Germany) with a 32-channel receive 1-channel transmit head coil (Nova Medical, Cambridge, MA, USA), at Beijing MRI center for Brain Research (BMCBR). The gradient coil has a maximum amplitude of 70 mT/m, 200 us minimum gradient rise time, and 200 T/m/s maximum slew rate. Functional images were acquired with a $T2^*$-weighted 2D GE-EPI sequence (1.5-mm in-plane resolution, 1.5-mm slice thickness without gap, 68 axial slices, TR = 2,000 ms, TE = 21.6 ms, flip angle = 80˚, image matrix = $122 \times 122$, FOV = $183 \times 183$ mm, partial Fourier factor = 6/8, bandwidth = 1,576 Hz/Px, GRAPPA acceleration factor = 2, phase encoding direction from A to P). A few EPI images with reversed phase encoding direction (P to A) were also acquired to correct image distortions in the phase encoding direction. Anatomical images were acquired with a T1-weighted MP2RAGE sequence (0.7-mm isotropic voxels, 256 sagittal slices, FOV = $224 \times 224$ mm, TR = 4,000 ms, TE = 3.05 ms, TI1 = 750 ms, flip angle = 4˚, TI2 = 2,500 ms, flip angle = 5˚, bandwidth = 240 Hz/Px, phase partial Fourier = 7/8, slice partial Fourier = 7/8, GRAPPA = 3). To improve data quality, a bite-bar was used to restrict head motion.

**MRI data preprocessing.** MRI data were analyzed using AFNI [59], FreeSurfer [60] (version 6.0), ANTs [61], and the lab-developed mripy package (https://github.com/herrlich10/mripy). The preprocessing of volumetric data includes slice timing correction, EPI image nonlinear distortion correction with reversed-blip method, linear motion correction, T1w anatomical image registration to EPI volumes, spatial normalization to MNI space, and per-run scaling to percent signal change before general linear model (GLM) analysis. For spatial normalization of subcortical areas, we estimated a 12-parameter linear transformation from the anatomical volume to a high-resolution MNI template (ICBM 152 2009c symmetric T1w template), followed by nonlinear transformation with ANTs (v2.1.0) using a subcortical mask focused on the areas of interest. All spatial transformations (including motion correction, anatomical to functional volume registration, and spatial normalizations) were combined altogether and applied to the functional volumes in one interpolation step (cubic method) at 0.6 mm isotropic resolution.

A surface-based approach was used for cortical data analysis. The T1w MP2RAGE anatomical volume was segmented into white matter, gray matter, and cerebrospinal fluid using FreeSurfer's automated procedure, using its high-resolution option. The preprocessed volumetric data before spatial normalization were mapped to the inflated cortical surface. Surface data were then normalized to a standard surface (std141) with surface-based warping, followed by surface-based smoothing with a 4.5-mm FWHM (full-width half-maximum) Gaussian kernel [62]. Surface data were then normalized to percent signal change before GLM analysis.

**General linear model (GLM) analysis.** BOLD signal changes from baseline for each stimulus condition were estimated using a GLM with fixed HRFs. For cortical data, we used a canonical HRF (BLOCK4 in AFNI). For subcortical data, we used a faster and narrower HRF that is more appropriate for subcortical regions [63]. Motion parameters, their derivatives and square of derivatives, were included as regressors of no interest. In addition, we found no covariation of motion parameters with different trial conditions.

**ROI definition and group-level statistical maps.** As in S8 Fig, ROIs for the SC, PBGN, LC, VTA, and LGN were carefully drawn on the MNI template according to their anatomical landmarks and atlases [64,65]. ROIs for the pulvinar subdivisions were obtained from a parcellation based on task-coactivation profiles [66]. The ROI of amygdala was defined based on a manually delineated high-resolution atlas of subcortical nuclei [67]. Before group-level analysis, spatial smoothing was performed on individual beta maps of GLM results *within* the ROI with a 1.4-mm FWHM Gaussian filter. Group-level statistical maps were then generated using standard random-effects analysis (*t* test) on the beta maps across participants. Cortical ROIs were defined on the cortical surface. ROIs of visual cortical areas were defined with a retinotopic atlas generated with the Human Connectome Project retinotopy dataset [68]. The anatomical masks for frontoparietal ROIs were defined on the standard surface by the HCP-MMP1 atlas [69] (S9 Fig).

**ROI-averaged responses.** For the subcortical ROI, we obtained the averaged response of all voxels in the anatomical ROI as defined by S8 Fig. To validate the collision sensitivity in a subcortical area with independent voxel selection, a LOSO cross-validation approach was used. For each participant, the ROI to calculate the collision-sensitive response was defined by voxels with group-level collision sensitivity (hit-miss, $p < 0.05$ uncorrected) from the remaining participants. For the visual cortical areas, we utilized an intraparticipant LORO 4-fold cross-validation approach to define the retinotopic ROIs. For each quadrant, we selected the visually responsive voxels with 3 runs of data (hit + miss $p < 0.05$ uncorrected) within the Benson14 retinotopic atlas [68]. Subsequently, BOLD responses from the left-out run from the retinotopic ROI were obtained. The procedure was repeated 4 times, and the results from all iterations were averaged for this quadrant. Finally, the results from all 4 quadrants were averaged as the ROI-averaged responses for the visual area.

**Correlation analysis of collision-sensitive responses.** To investigate the relationship of collision sensitivity between different brain areas, we calculated Pearson's correlations between the hit-miss responses in subcortical and cortical areas across participants. To avoid the risk of circular testing, collision-sensitive responses were determined by the LORO 4-fold cross-validation approach. In the attended or unattended condition, collision-sensitive voxels were selected by 3 runs sorted by the hit-miss T statistics, and the ROI averaged hit-miss responses from these voxels were extracted from the remaining run. The procedure was repeated for each run, and the results were averaged as the final estimate. Considering that the average size of collision-sensitive clusters in subcortical nuclei (SC, vmPul, and VTA) is about 30 μl, 10 voxels (1.5 mm isotropic) with the strongest collision sensitivity were selected for each subcortical ROI. For the visual cortex, 50 voxels with the strongest collision sensitivity were selected for each quadrant (200 voxels in total). For the attentional network, 200 voxels with the strongest collision sensitivity were selected.

**Beta-series functional connectivity.** A beta-series method was adopted to test whether functional connectivity existed between the SC and vmPul [34]. The response for each trial was estimated with a separate regressor in a GLM analysis of the ROI-averaged BOLD time series from collision sensitive voxels, generating a series of beta values for each ROI. The collision sensitive voxels for each ROI were defined by the LORO approach described from above. The beta series for each trajectory condition were Z-scored to avoid the influence of mean amplitude differences. A second-level linear regression was conducted with any 2 series of these beta-values from 2 different ROIs, and the second-level regression coefficient was obtained as the functional connectivity.

**Path analysis by structural equation modeling.** Path analysis under SEM framework was performed to infer the causal influence of beta series between subcortical regions (SC, vmPul, and VTA), visual cortex (including V1, V2, V3, V3a, V3b, and TO1/TO2) and frontoparietal

attention networks (S8 Fig). The collision-sensitive responses across all participants were concatenated as the model input. A total of 216 possible models were constructed by all combinations of 6 groups of possible alternative connections. By doing so, we could directly compare the strength and possibility of our alternative hypotheses. In addition, several common connections were added in all models based on the prior knowledge. All path coefficients in each model were freely determined by maximum likelihood estimate in the SEM with lavaan software 0.6–3 [70]. The model selection was based on several indices. We first excluded models with a parsimony goodness of fit index (PGFI) larger than 0.1, then ranked the remaining models by their adjusted goodness of fit index (AGFI), and, finally, picked the best fitted models for the attended and unattended conditions [71]. Other goodness of fit indices including $\chi^2$, comparative fit indices (CFIs), goodness of fit index (GFI), root mean square residual (RMR), and root mean square error of approximation (RMSEA) were also calculated and ranked to ensure the best model was not sensitive to the ranking methods. Different ranking methods led to the same results.

### Experiment 3: fMRI study with hemianopic patients

**Participants.**   A total of 12 hemianopic patients (23 to 64 years old, mean age = 45.5 years, SD = 14.1 years, 1 female) with unilateral lesions of the geniculostriate pathway and no other neurological or psychiatric conditions were enrolled from General Hospital of People's Liberation Army, Beijing, China. Patients lost their conscious vision in one-half or a quadrant of the visual field from both eyes. They had normal or corrected-to-normal vision outside the BVF. Clinical characteristics of all patients were shown in S1 Table. Their Humphrey perimetry and lesioned locations were shown in S11 Fig. The scotoma was defined as the visual field with a relative sensitivity less than −20 dB and $p < 0.5\%$ compared with normal population in both eyes.

**Stimuli and procedures.**   Visual stimuli and procedures were similar as those in experiment 2. Looming trajectories were slightly changed (S1C Fig), moving from 10.75 meters to 2.75 meters in front of the observers. In addition to the "hit" trajectory (hit the eye) and "near-miss" trajectory (passing at 5 cm of horizontal shift from the eye), a "receding" trajectory was also included with the reverse trajectory as in the "hit" condition. For the 7T experiment, the on-screen size of the object changed between 0.32 and 1.25 degrees of visual angle, at about 3.99 degrees of eccentricity (angle with vertical meridian is 60˚). For the 3T experiment, the on-screen size of the object changed between 0.45 and 1.75 degrees of visual angle, at about 5.58 degrees of eccentricity (angle with vertical meridian is 60˚). Compared with the stimulus in experiment 2, the looming stimulus was smaller and disappeared earlier (time-to-collision at the vanishing point was 115 ms). Note that we adjusted the brightness of the looming object and the background between the three experiments. In experiments 1 and 3, the object was brighter than the background, while in experiment 2, it was made darker. Stimuli were presented either in the NVF or in the BVF of hemianopic patients. In principle, the stimulus location was selected based on the scotoma of each patient. Stimuli were presented at the lower visual field for P02 and P08 and at the upper visual field for other patients. For P11, stimuli were presented in the upper quadrant of the BVF, while in the lower quadrant of the NVF because he could see better than in the upper quadrant. For P12, visual stimuli were presented at an eccentricity of 7.84 degrees and much closer to the vertical meridian (angle with vertical meridian is 30˚). Since the stimulus location of P12 was very different from those of other participants, we did not use his data to generate the group-averaged results in Fig 6.

During fMRI scans, patients were asked to keep fixation and to detect occasional color changes of the fixation point. For each patient, 6 runs were collected, except 5 runs for P01 and

4 runs for P03. Each run consisted of 24 trials in which each trajectory was repeated 4 times in the NVF and 4 times in the BVF. The ISI was randomly selected from 8, 10, 12, and 14 seconds.

Four patients (P08 to P12) also performed a behavioral test of stimulus visibility in their BVF. In the subjective visibility test, they were asked to keep fixation and report the visibility to stimuli presented to the NVF or BVF. In the 2-IFC test, the stimulus was presented in one of two 330-ms intervals, separated by an 833-ms gap in-between. The first interval was accompanied by a low-pitch tone at 300 Hz and the second by a high-pitch tone at 700 Hz. Patients were required to keep fixation and to report in which interval the object was presented. Auditory feedback was given after incorrect answers. For each patient, 240 trials were collected, including 24 trials in the NVF (8 trials for each trajectory) and 216 trials in the BVF (72 trials for each trajectory). In the behavioral test, the looming stimulus was very close to that in the 7T experiment, with an on-screen size expanding approximately from 0.31 to 1.21 degrees at 3.95 degrees of eccentricity. For P12, the stimulus was at 7.84 degrees of eccentricity.

**MRI data acquisition and analysis.** MRI data for P01, P04, P05, P07, P10, and P11 were acquired with the same 7T scanner, head coil, and pulse sequences as in experiment 2. For P01, functional images were acquired with GE-EPI at a higher spatial resolution (1.2-mm isotropic voxels, 62 axial slices of 1.2-mm thickness, $150 \times 150$ matrix, TR/TE = 2,000/22 ms, nominal flip angle = 78˚, partial Fourier factor = 6/8, GRAPPA acceleration factor = 2, multiband factor = 2, bandwidth = 1587 Hz/Px, phase encoding direction from A to P).

P02, P03, P06, P08, P09, and P12 had nonmagnetic metal implants. Due to safety issues of overheating at ultrahigh magnetic fields, their data were acquired with a 3T scanner (Siemens Prisma, Erlangen, Germany) using a 20-channel phased array coil (Nova Medical, Cambridge, MA, USA). High-resolution anatomical images were acquired using a T1-weighted MPRAGE sequence (1-mm isotropic voxels, 192 sagittal slices at 1-mm thickness, image matrix = $256 \times 224$, TR/TE = 2,600/3.02 ms, inversion time = 900 ms, flip angle = 8˚, bandwidth = 130 Hz/Px, phase partial Fourier = 6/8, slice partial Fourier = 7/8, no in-plane acceleration). Functional images were acquired with a GE-EPI sequence (2-mm isotropic voxels, 52 or 54 axial slices of 2-mm thickness, $96 \times 96$ matrix, FOV = $192 \times 192$ mm, TR = 2,000 ms, TE = 30 or 31.4 ms, flip angle = 80˚, multiband or SMS factor = 2, partial Fourier factor = 7/8 or none, bandwidth = 1,860 or 2,170 Hz/Px, phase encoding direction from A to P, no in-plane acceleration). A few EPI images with reversed phase encoding direction (P to A) were acquired to correct image distortions in the phase encoding direction.

Data analysis procedures were the same as those in experiment 2, except for cortical data analysis. A 6-mm FWHM spatial smoothing was performed on functional volumes after motion correction, followed by per-run scaling and GLM analysis. Statistical volumes were spatially normalized to the MNI space with ICBM 152 symmetric MNI template (2009c) with a combination of linear and nonlinear transformations. Group-level statistics were generated in the MNI space and then projected to the standard cortical surface (std141) as shown in Fig 6B.

Since the stimulus location of P12 was very different from those of other participants, we did not use his data to generate the group-averaged results for the subcortical area with strong retinotopic representations, including the SC, pulvinar, and LGN. But P12 was included in the individual results in Fig 6I and 6J. P03 had severe damage and distortion in his ipsilesional visual thalamus, thus was not included in the results for ipsilesional pulvinar and LGN. Due to a similar reason, P09 was not included in the analysis of ipsilesional LGN. Therefore, for the group-level analysis, there were 11 contra- and ipsilesional SCs, 11 contralesional pulvinars and 10 ipsilesional pulvinars, 11 contralesional LGNs and 9 ipsilesional LGNs, and 12 contra- and ipsilesional VTAs.

## Statistical analysis

We adopted repeated measurements and two-sided design for all statistical tests in the study. Effect sizes including Cohen's $d$, $\eta_p^2$, and Pearson's r were computed via JASP (0.16, The JASP Team, https://jasp-stats.org). Cohen's d for $t$ tests was computed as M_diff / SD_diff, where M_diff denotes the mean of paired differences and SD_diff denotes the standard deviation of paired differences. $\eta_p^2$ for ANOVA was calculated as SS_effect / (SS_effect + SS_error), where SS_effect denotes the sum of squares for the within-participants effect and SS_error denotes the sum of squares for the error term. Other details about the statistical analysis were described as below.

## Behavioral statistics

A permutation test (exact test, in which all possible permutations were considered) was used to infer the significance of difference between discrimination sensitivities in the upper and lower visual field in Fig 1C. A cluster-based permutation test (10,000 times of permutations) was used to calculate the significant time periods in Fig 1D. The length of continuous periods (i.e., clusters) of time points showing significant difference in each permutation was used to control the FWE. Paired $t$ tests were used to calculate the difference in pupil diameters between different trajectories in Fig 1E and 1F. The difference of behavioral performance of hemianopic patients against chance-level and between trajectories were assessed using a binomial generalized linear mixed model with a logit link function. In the model, we tested a fixed effect of trajectory while controlling for a random effect of participants, with likelihood ratio tests method. The linear relationship of SC responses with the behavioral performance was examined via a linear mixed model analysis, with random effect of participants.

## Statistical maps of fMRI

Standard random-effects analysis was used to generate the statistical maps of subcortical and cortical areas. To control the FWE of statistical maps within each subcortical ROI (small volume correction), a cluster mass–based permutation test was used to calculate the $p$-value of significant clusters [72]. The cluster mass was defined as the absolute sum of T values in a cluster. The cluster defining threshold is voxel's $p < 0.05$. In each permutation, the hit and miss conditions were randomly switched for each participant, followed by a group-level $t$ test across participants to generate the statistical map. The largest cluster mass was then recorded for this permutation. The permutation procedure was repeated 10,000 times to get the null distribution for the largest cluster mass. A cluster $p$-value was derived as the proportion of the null distribution larger than the actual cluster mass. To obtain an accurate null distribution, we subtracted the group-averaged map from each individual map (demean) and removed permutations less than 15% or 85% of data exchanges. FWEs of surface clusters (at $p < 0.05$ for individual vertex) were determined by Monte Carlo simulation in AFNI.

**Statistical analysis of ROI-averaged responses.** Statistical analysis of ROI-averaged responses in experiment 2 was performed with two-way repeated measures ANOVA. Sphericity was validated by Mauchly's test, and normality by Shapiro–Wilk test. To control FWEs, we followed the Fisher logic [73] and only performed post hoc $t$ tests when there was a significant main effect or interaction of the two-way ANOVA. Associations between variables were assessed with Pearson correlations after removing outliers outside 1.5 times interquartile ranges of robust Mahalanobis distances of all samples (calculated using MATLAB function robustcov based on the FAST-MCD method). The FDRs of collision sensitivity across ROIs in Fig 4A were calculated by the Benjamini–Hochberg method. The FWEs of correlations in Fig 5A were calculated by a permutation test. The correspondences of data pairs were randomly

shuffled in each permutation. The largest absolute value of correlation coefficients was then recorded to compose the null distribution. The corrected $p$-values were derived from the null distribution for the largest correlation coefficient. An LME model was used to analyze ROI-averaged responses from visually responsive voxels in experiment 3. Trajectory, visual field, and ROI were defined as the fixed effect, and participant as the random effect.

**Multiple tests correction across subcortical ROIs.** As we performed multiple tests of collision sensitivity in several subcortical ROIs in experiment 2, a sequential Holm–Bonferroni approach was used to correct FWEs. The $p$-values of collision sensitivity in the foveal SC, vmPul, VTA, PBGN, LC, and amygdala were included for correction. Holm–Bonferroni correction was first performed within each ROI to adjust the $p$-values for the collision sensitivity of ROI-averaged responses, the collision-sensitive cluster, and the LOSO result. For the SC, the ROI-averaged response and LOSO result in the unattended condition still show significant collision sensitivity after Holm correction ($p = 0.021$ and $0.024$, respectively). Since we have a well-founded hypothesis to test the collision sensitivity in the SC, it was not included in subsequent multiple-test correction across ROIs. For the other downstream subcortical ROIs, the lowest $p$-value after correction within each ROI was selected for correction. After Holm correction, the vmPul and VTA still showed a significant collision sensitivity in the unattended condition ($p = 0.03$ and $0.032$, respectively).

## Supporting information

**S1 Fig. Top view of stimulus trajectories. (a)** Experiment 1. A baseball-sized sphere launched 11.3 meters away from the observer, moving at a speed of 24 m/s. It disappeared at 3.3 meters from the observers at 138 ms of time-to-collision, as indicated by the location of the arrows. Yellow, red, blue, and green arrows indicate the hit-nasion, hit-eye, near-miss, and far-miss trajectories. Stimuli were presented on a 3D monitor (width = 0.51 m) 1.3 meters in front of the observers. The on-screen display was a sphere expanding from 0.3 to 1 degree of visual angle in 330 ms. While the vertical offset of the stimulus from the center of screen was 1.90˚, the horizonal offset for the starting (hit nasion: 3.14˚, hit eye: 2.79˚, near miss: 2.43˚, far miss: 2.13˚) and ending (hit nasion: 2.77˚, hit eye: 2.79˚, near miss: 2.80˚, far miss: 2.86˚) position of the stimulus varied between different trajectory conditions. **(b)** Experiment 2. Stimuli were presented on a translucent screen with a 2D projector. The sphere moved from 8.75 m away and disappeared on the screen (width = 0.35 m) 0.75 m in front of the observer at 33 ms of time-to-collision. Red and blue arrows indicate the hit and near-miss trajectories. The stimulus on the screen expanded from 0.4 to 4.5 degrees of visual angle in 330 ms. While the vertical offset of the stimulus from the center of screen was 2.49˚, the horizonal offset for the starting (hit: 4.31˚, near miss: 1.70˚) and ending (hit: 4.31˚, near miss: 5.87˚) position of the stimulus varied between different trajectory conditions. **(c)** Experiment 3. In hit and near-miss conditions, the sphere moved from 10.75 m to 2.75 m in front of the observer. The time-to-collision at disappearance was 115 ms. The receding trajectory was the reverse of the hit trajectory. Red, blue, and green arrows indicate the hit, near-miss, and receding trajectories. The eccentricity and the size of the sphere (7T: 6 cm, screen width = 0.35 m; 3T: 8.4 cm, screen width = 0.51 m) was slightly different in the 7T and 3T scanning. For the 7T (or 3T) experiment, the on-screen size of the object changed between 0.32 and 1.25 (3T: 0.45 and 1.75) degrees of visual angle. While the vertical offset of the stimulus from the center of screen was (7T: 2˚; 3T: 2.79˚), the horizonal offset for the starting (7T: hit: 3.46˚, near miss: 2.82˚; 3T: hit: 4.83˚, near miss: 3.95˚) and ending (7T: hit: 3.46˚, near miss: 3.60˚; 3T: hit: 4.83, near miss: 5.04) position of the stimulus varied between different trajectory conditions.
(PDF)

**S2 Fig. Overall magnitude of optical flows and luminance change in hit and near-miss looming stimuli (experiment 2).** We used the Horn–Schunck method [74] to compute the frame-by-frame optical flow of our stimuli. The Horn–Schunck method is utilized to determine the optical flow by analyzing the displacement of pixels between frames, assuming that the brightness of a pixel remains constant during its motion. In our analysis, we captured the image in each frame during the visual stimulation. Subsequently, we employed the MATLAB Optical Flow block (the mathematical algorithm [74] is given in S1 Text) from the Computer Vision Toolbox to estimate the optical flow vector for each pixel between 2 consecutive frames. The direction of pixel motion was determined by the horizontal and vertical components of the vector, while the speed or magnitude was calculated as the square of the vector's modulus. Finally, we computed the overall flow magnitude by summing the magnitudes of all the pixels. The scripts for this analysis have been uploaded to https://doi.org/10.5281/zenodo.8251435. **(a)** The original stimulus images were overlaid with red lines depicting the resulting optical flow. Three example frames were displayed for each condition. **(b)** Upon comparing the magnitudes of the optical flows, it was observed that the near-miss stimulus exhibited a slightly larger overall optical flow compared to the hit stimulus. **(c)** To further analyze the data, we plotted the change in optical flow magnitude at various eccentricities, each shown in different panels. **(d)** Additionally, we calculated the time-integrated optical flow at different eccentricities. The resulting figure illustrated that the near-miss stimulus generated a greater optical flow in the central visual field when compared to the hit stimulus. Therefore, the collision-sensitive activations in the foveal SC cannot be accounted by a stronger optical flow in the fovea. **(e)** Similarly, we also obtained the time-integrated luminance change (from the background) for both the hit and near-miss stimuli at different eccentricities. This was accomplished by calculating the light intensity of each pixel in each frame using spectroradiometer data. It was observed that the near-miss stimulus produced a greater decrease in luminance at lower eccentricities. Thus, the difference in luminance changes cannot account for the collision-sensitive activations in the foveal SC.
(PDF)

**S3 Fig. V1 response as a function of eccentricity (experiment 2).** To further support that the stimuli have no systematic bias in retinotopic location, we plotted V1 response profiles as a function of eccentricity. V1 vertices corresponding to the polar angle of the visual stimuli from 0 to 10 degrees of eccentricity were selected based on the HCP retinotopic atlas [68,75]. As shown in the figure below, there was no difference in foveal activations in V1 between the hit and miss stimulus conditions. Therefore, our findings of collision-sensitive activations in the SC cannot be explained by a foveal retinotopic bias to the hit stimulus.
(PDF)

**S4 Fig. (a) Individual data for the behavioral experiment (experiment 1).** The percentage of hit responses to looming stimuli as a function of impact points was fitted with a normal CDF. In experiment 1, the horizontal offset of the would-be impact point for the near miss and far miss conditions was individually adjusted according to the performance of each participant in a preliminary session. The would-be impact point for the near miss and far miss conditions were chosen at approximately 50% and 2% of hit response, respectively. **(b) Distribution of eye positions for hit and miss looming stimuli from four quadrants of the visual field.** The eye gaze positions from stimulus onset to 1,000 ms after were analyzed. Error bars indicate 3′ standard deviation of the distribution. No significant difference was found between the eye positions to the hit and miss stimuli.
(PDF)

**S5 Fig. Pupil size changes to looming stimuli in a bright background. (a)** Stimulus diagram was the same as in experiment 1, but with a brighter background than looming stimuli. A total of 10 participants (7 females and 3 males) participated in this experiment. Participants performed a collision detection task, i.e., paying attention to the approaching object. In addition, there was a task-irrelevant rapid presentation of letter streams in the fixation, similar to the fixation change in the unattended condition in experiments 2 and 3 (note in these experiments the fixation change was task relevant). **(b, c, d)** Time courses of the pupil size change for upper +lower visual field (VF) data, upper VF data only, and lower VF data only, respectively. The light green bar at the bottom indicates the uncorrected significant difference between hit and miss conditions. No significant difference was found after multiple comparison corrections via permutation. **(e, f)** Bar plots of the pupil size during the looming component (see Fig 1D) for the upper VF only data and lower VF only data, respectively. No significant difference was found between the bars.
(PDF)

**S6 Fig. Pupil size changes to looming stimuli in the unattended condition with a bright background. (a)** Stimulus diagram was the same as in experiment 2, but with a slightly longer viewing distance (0.85 m). A total of 17 participants (8 females and 9 males) participated in this experiment. Participants were instructed to count the number of color changes of central fixation point. **(b, c)** The time courses of changes in pupil size for stimuli in the upper **(b)** and lower **(c)** visual field are presented. The light green bar at the bottom indicates the uncorrected significant difference between hit and miss conditions. No significant difference was found after multiple comparison corrections via permutation.
(PDF)

**S7 Fig. Looming-evoked responses across the SC (experiment 2).** The first row of each panel shows the retinotopic activations with significantly stronger responses to contralateral than to ipsilateral stimuli. Red lines on the sagittal view indicate the location of the coronal slices. The second to the fourth rows show the activation maps for the hit, miss, and hit-miss responses. Maps were thresholded at voxel $p < 0.05$ uncorrected.
(PDF)

**S8 Fig. Anatomical ROIs of subcortical nuclei (experiment 2 and experiment 3).** From left to right are the superior colliculus (SC, 261 μl), ventral tegmental area (VTA, 708 μl), parabigeminal nucleus (PBGN, 90 μl), locus coeruleus (LC, 86 μl), amygdala (1883 μl), lateral geniculate nucleus (LGN, 252 μl), and pulvinar. The pulvinar was parcellated into 5 subdivisions based on the task-coactivation patterns [66], including the ventromedial pulvinar (vmPul, red, 284 μl), ventrolateral pulvinar (vlPul, orange, 359 μl), dorsolateral pulvinar (dlPul, yellow, 246 μl), dorsomedial pulvinar (dmPul, green, 224 μl), and anterior pulvinar (aPul, blue, 215 μl). The pulvinar ROIs used in this study were defined as the intersections of the original ROIs in the left and right hemispheres.
(PDF)

**S9 Fig. Anatomical ROIs for frontoparietal attention networks (experiment 2).** ROIs for dorsal (dAN) and ventral (vAN) attention networks were defined based on anatomical landmarks and HCP-MMP1 atlas. Areas of dAN include IPS/SPL (IPS1, MIP, VIP, LIPv, LIPd, IP1, and 7PL) and SFC (6a and FEF). Areas of vAN include TPJ (TPOJ1, STV, PSL, and PF) and IFC (PEF, IFJb, IFJa, IFSp, and 6r). Yellow annotations indicate selected ROIs in HCP-MMP1 atlas.
(PDF)

**S10 Fig. Activation maps and ROI-averaged responses to looming stimuli in other subcortical nuclei (experiment 2). (a, c)** LGN. **(b, d)** Amygdala. **(e, g)** LC. **(f, h)** PBGN. Statistical maps were thresholded at $p < 0.05$ uncorrected. No significant collision-sensitive cluster or ROI-averaged response can be found from these areas. LOSO analysis revealed a significant collision sensitivity in the PBGN in the unattended condition ($p < 0.025$), but it cannot survive the correction across multiple tests.
(PDF)

**S11 Fig. Clinical perimetry, lesioned locations, and stimulus-evoked occipital activations for hemianopic patients (experiment 3).** For each patient, the left panels show the Humphrey perimetry of visual field test. In the structural image below, relevant lesioned locations were indicated by red dashed ovals. For P17, both T1w and diffusion tensor images were shown to indicate the lesion of right optic radiation. In the middle panels, the scotoma was depicted schematically within 10 degrees of eccentricity in black color (i.e., relative sensitivity $< -20$ dB and $p < 0.5\%$ compared with normal population), with the yellow sphere indicating the stimulus in the fMRI experiment. The right panels show the occipital activations to stimuli presented to the NVF and BVF (indicated by red arrows). Although clear contralateral V1 activations can be observed to stimuli presented to the NVF, most patients (8/12) showed no significant V1 activation in the lesioned hemisphere to stimulus presented to the BVF (the left bar graph below: dashed line for individual data, *** for $p < 0.001$). For the 4 patients (P02, P04, P06, and P08) showing weak uncorrected activations in the occipital lobe of the lesioned hemisphere, no significant difference was found in the responses of these voxels to the hit and miss stimuli (the right bar graph below), which cannot explain the collision-sensitive responses in the SC (Fig 6A).
(PDF)

**S12 Fig. Looming-evoked responses in the LGN of hemianopic patients (experiment 3).** Group-averaged activation maps were thresholded at $p < 0.05$ uncorrected. Red arrows indicate the location of visually evoked response to receding stimuli in the LGN. Blue dotted lines denote the anatomical boundary of the LGN. No collision sensitivity was found from the ROI-averaged response of the whole LGN, nor from the LOSO analysis.
(PDF)

**S1 Table. Clinical characteristics of hemianopic patients.**
(PDF)

**S1 Text. Equations to calculate optical flow.**
(PDF)

## Author Contributions

**Conceptualization:** Jinyou Zou, Sheng He, Peng Zhang.

**Data curation:** Fanhua Guo, Jinyou Zou, Ye Wang, Peng Zhang.

**Formal analysis:** Fanhua Guo, Jinyou Zou, Ye Wang, Peng Zhang.

**Funding acquisition:** Peng Zhang.

**Investigation:** Jinyou Zou, Sheng He, Peng Zhang.

**Methodology:** Fanhua Guo, Jinyou Zou, Ye Wang, Peng Zhang.

**Project administration:** Peng Zhang.

**Resources:** Boyan Fang, Huanfen Zhou, Dajiang Wang, Peng Zhang.

**Software:** Fanhua Guo, Jinyou Zou, Ye Wang, Peng Zhang.

**Supervision:** Jinyou Zou, Dajiang Wang, Sheng He, Peng Zhang.

**Validation:** Fanhua Guo, Jinyou Zou, Ye Wang, Peng Zhang.

**Visualization:** Fanhua Guo, Jinyou Zou, Ye Wang, Peng Zhang.

**Writing – original draft:** Fanhua Guo, Jinyou Zou, Peng Zhang.

**Writing – review & editing:** Fanhua Guo, Jinyou Zou, Ye Wang, Sheng He, Peng Zhang.

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
