## [Editor Report · Decision Letter 0]

11 Oct 2023

Dear Dr Zhang, 

Thank you for submitting your manuscript entitled "Human subcortical pathways automatically detect collision trajectory without attention and awareness" for consideration as a Research Article by PLOS Biology.

Your manuscript has now been evaluated by the PLOS Biology editorial staff as well as by an academic editor with relevant expertise and I am writing to let you know that we would like to send your submission out for external peer review.

Once your full submission is complete, your paper will undergo a series of checks in preparation for peer review. After your manuscript has passed the checks it will be sent out for review. To provide the metadata for your submission, please Login to Editorial Manager (https://www.editorialmanager.com/pbiology) within two working days, i.e. by Oct 13 2023 11:59PM.

Kind regards,

Christian

Christian Schnell, PhD

Senior Editor

PLOS Biology

cschnell@plos.org

---

## [Editor Report · Decision Letter 1]

20 Nov 2023

Dear Peng,

Thank you for your patience while we considered your revised manuscript "Human subcortical pathways automatically detect collision trajectory without attention and awareness" for publication as a Research Article at PLOS Biology. This revised version of your manuscript has been evaluated by the PLOS Biology editors and the Academic Editor.

Based on the reviews and on our Academic Editor's assessment of your revision, we are likely to accept this manuscript for publication, provided you satisfactorily address the remaining points raised by the Academic Editor. Please also make sure to address the following data and other policy-related requests.

* Please include some of the figures from the last rebuttal as supplementary figures, in particular the eye movement analysis and the V1 responses as a function of eccentricity were helpful for assessing the manuscript.

* ETHICS STATEMENT: Please also include an approval number.

* DATA POLICY:

1C, 1E, 1F, 2C, 2D, 2E, 2F, 3B, 3D, 4A, 5B, 6E, 6G, 6H, and similar graphs in the supplementary information

* CODE POLICY

Per journal policy, as the code that you have generated is important to support the conclusions of your manuscript, we require that you make it available without restrictions upon publication. Please ensure that the code is sufficiently well documented and reusable, and that your Data Statement in the Editorial Manager submission system accurately describes where your code can be found. We could find the zenodo repository that you refer to. Please make sure that it is publicly available.

* Financial disclosure: Please include links to the funders in the "Financial Disclosure" statement.

* Please provide an editorial blurb.

We expect to receive your revised manuscript within two weeks. 

*Published Peer Review History*

*Press*

Sincerely,

Christian

Christian Schnell, PhD

Senior Editor,

cschnell@plos.org,

PLOS Biology

---

## [Editor Report · Decision Letter 2]

8 Dec 2023

Dear Dr Zhang,

Thank you for your patience while we considered your revised manuscript "Human subcortical pathways automatically detect collision trajectory without attention and awareness" for publication as a Research Article at PLOS Biology. This revised version of your manuscript has been evaluated by the PLOS Biology editors.

We are likely to accept this manuscript for publication but need you to address the following remaining concerns: 

* Please provide links to the web sites of the funding organizations that you list in the Financial Disclosure section of the manuscript details. 

* Thank you for uploading your data and code to OSF. However, the data for the individual figures are not easy to find. Could you please add an excel or csv file that includes the data points underlying the summarizing figures (1C, 1E, 1F, 2C, 2D, 2E, 2F, 3B, 3D, 4A, 5B, 6E, 6G, 6H). You can either provide one file for each panel/figure or one file and different tabs for each figure/panel. 

* Please also add to the figure legends of the corresponding figures where the data underlying the figures can be found. For example: "Data underlying the graphs in this figure can be found at https://osf.io/gdjwh/"

We expect to receive your revised manuscript within two weeks. 

*Published Peer Review History*

*Press*

Sincerely,

Christian

Christian Schnell, PhD

Senior Editor

cschnell@plos.org

PLOS Biology

---

## [Editor Report · Decision Letter 3]

14 Dec 2023

Dear Dr Zhang,

Thank you for the submission of your revised Research Article "Human subcortical pathways automatically detect collision trajectory without attention and awareness" for publication in PLOS Biology. On behalf of my colleagues and the Academic Editor, Christopher Pack, I am pleased to say that we can in principle accept your manuscript for publication, provided you address any remaining formatting and reporting issues. These will be detailed in an email you should receive within 2-3 business days from our colleagues in the journal operations team; no action is required from you until then. Please note that we will not be able to formally accept your manuscript and schedule it for publication until you have completed any requested changes.

PRESS

Sincerely, 

Christian

Christian Schnell, PhD, PhD

Senior Editor

PLOS Biology

cschnell@plos.org